# Beyond Token-level Supervision: Unlocking the Potential of Decoding-based Regression via Reinforcement Learning

**Ming Chen** [* 1 2]   **Sheng Tang** [* 1 2]   **Rong-Xi Tan** [* 1 2]   **Ziniu Li** [3]   **Jiacheng Chen** [4]   **Ke Xue** [† 1 2]   **Chao Qian** [† 1 2]

## Abstract

Decoding-based regression, which reformulates regression as a sequence generation task, has emerged as a promising paradigm of applying large language models for numerical prediction. However, its progress is hindered by the misalignment between discrete token-level objectives (e.g., cross-entropy) and continuous numerical values. Existing approaches relying on token-level constraints often fail to capture the global magnitude of the target value, limiting their precision and generalization. In this paper, we propose to unlock the potential of decoding-based regression via reinforcement learning. We formulate the generation process as a Markov decision process, utilizing sequence-level rewards to enforce global numerical coherence. Under this framework, we present GenRe$^2$, which combines policy gradient methods and on-policy distillation to provide dense expert supervision while preserving error magnitudes, thereby resolving the temporal credit assignment challenge. Extensive experiments across tabular regression, code metric prediction and generative reward modeling demonstrate that GenRe$^2$ consistently outperforms traditional baselines, establishing a robust paradigm for general-purpose numerical prediction.

## 1. Introduction

Regression, the task of predicting continuous targets from input representations, stands as a fundamental role of machine learning (Bishop, 2006; Van Breugel & Van Der Schaar,

---
[*]Equal contribution [1]State Key Laboratory of Novel Software Technology, Nanjing University [2]School of Artificial Intelligence, Nanjing University [3]School of Data Science, The Chinese University of Hong Kong, Shenzhen [4]Department of Computer Science and Engineering, The Chinese University of Hong Kong. Correspondence to: Ke Xue <xuek@lamda.nju.edu.cn>, Chao Qian <qianc@lamda.nju.edu.cn>.

*Proceedings of the 43rd International Conference on Machine Learning*, Seoul, South Korea. PMLR 306, 2026. Copyright 2026 by the author(s).

2024), with wide applications across critical domains ranging from scientific discovery (Hu et al., 2024) to industrial scenarios (He et al., 2025). Traditional regression methods, including Gaussian processes (Rasmussen & Williams, 2006) and tree-based models (Chen & Guestrin, 2016; Prokhorenkova et al., 2018), excel due to their robustness and interpretability (Sahakyan et al., 2021). However, with the advent of the Deep Learning (DL) era and the increasing complexity of data, there has been a paradigm shift towards DL-based regressors (Ye et al., 2024a; Jiang et al., 2026). These methods leverage the power of representation learning to map high-dimensional inputs into latent spaces, subsequently modeling the target value through specialized regression heads.

For DL-based regressors, there have been some design philosophies of regression heads to map latent representations to continuous targets. The most common approach, the pointwise head, projects representations directly to a scalar but often fails to capture the uncertainty or the complex multimodality of the target distribution (Lakshminarayanan et al., 2017). To address this, parametric distribution heads model outputs as predefined distributions (e.g., Gaussian), yet they rely on rigid assumptions that may not hold in real-world scenarios (Imani & White, 2018). Alternatively, the Riemann head (or histogram head) discretizes the continuous output into finite bins, converting regression into classification (Bellemare et al., 2017; Imani & White, 2018), showing great robustness (Imani et al., 2024) and performance (Müller et al., 2022). However, these methods primarily operate on structured data, limiting their ability to perform regression on the vast and diverse spectrum of unstructured data (e.g., text or code).

This limitation has motivated recent studies to leverage Large Language Models (LLMs) for universal regression (Vacareanu et al., 2024; Song et al., 2024; Tchuindjo & Khattab, 2025). A key development in this line of work is decoding-based regression (Song & Bahri, 2025), which reformulates regression as a discrete sequence generation task and can be trained over large amounts of regression data $(\mathbf{x}, y)$ represented as text. As illustrated in Figure 1, this approach reformulates regression as a next-token prediction task by tokenizing continuous values (e.g., via base-$B$

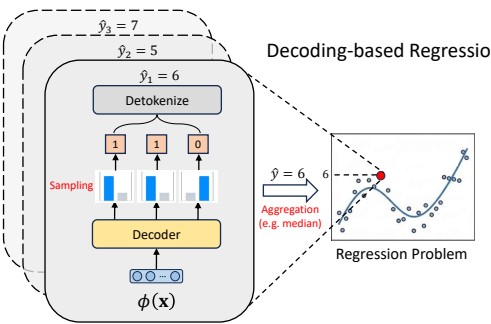

*Figure 1.* Illustration of decoding-based regression. The input **x** passes through an encoder to produce the representation $\phi(\mathbf{x})$, which is then processed by a decoder. The model performs multiple sampling trials to generate several discrete token sequences (e.g., the binary representation <1><1><0>). These sequences are individually detokenized into corresponding scalar values (shown in the stacked layers as $\hat{y}_1 = 6, \hat{y}_2 = 5, \hat{y}_3 = 7$). Finally, these scalar values are combined via an aggregation strategy (e.g., median) to produce the final prediction $\hat{y} = 6$.

expansion). Unlike traditional scalar regressors, decoding-based regression not only can handle unstructured raw data, but also leverages the strong sequential modeling capabilities of Transformers to capture complex distributions (Song et al., 2024). Furthermore, the generative approach of decoding-based regression mitigates the susceptibility to reward hacking often seen in scalar or histogram baselines (Chen et al., 2024; Yu et al., 2025), producing more robust and calibrated predictions, which align with the recent observations from generative reward models (Mahan et al., 2024; Zhang et al., 2025d). The concept of decoding-based regression gives rise to Regression Language Model (RLM) (Song et al., 2024), which demonstrates great potential in diverse applications ranging from industrial prediction (Akhauri et al., 2025; 2026) to black-box optimization (Nguyen et al., 2024; Tan et al., 2025a).

However, despite its promise, the potential of decoding-based regression remains locked. The critical barrier lies in the misalignment between the widely used Cross-Entropy (CE) loss and the numerical nature of the regression task (Lukasik et al., 2025). CE treats tokens as independent categories, ignoring their ordinal value and the entire magnitude of the detokenized number. While recent works have attempted to mitigate this via token-level distance penalties, e.g., NTL (Zausinger et al., 2025) and $\text{DIST}^2$ (Chung et al., 2026), a fundamental limitation remains: these methods operate locally on individual tokens and overlook the cumulative error over the entire sequence (Selvam, 2025), which can lead to catastrophic outcomes in the original numerical space (Song et al., 2024; Song & Bahri, 2025). Thus, there is an urgent need for a method that is inherently aware of sequence-level numerical magnitude.

In this paper, we propose **Gen**erative **Re**inforced **Re**gressor (GenRe$^2$) to bridge this gap. We reformulate decoding-

based regression as a Markov Decision Process (MDP), allowing us to optimize the model using policy gradient methods (Sutton et al., 1999). Unlike previous approaches, GenRe$^2$ utilizes a sequence-level reward signal, which is computed only after the full numerical sequence is generated and detokenized, to directly guide the model towards minimizing the true regression error (e.g., MSE). Crucially, recognizing the challenge of credit assignment in sparse-reward Reinforcement Learning (RL), we further integrate dense token-level On-Policy Distillation (OPD) into our framework. Motivated by the efficacy of hybrid reinforcement and imitation learning (Li et al., 2025a), GenRe$^2$ employs an expert oracle based on the ground-truth target to provide immediate feedback on the model's self-generated prefix. Here OPD provides step-wise guidance that sparse RL reward signals lack, while RL only leverages global signals to directly optimize the final regression metrics. To implement GenRe$^2$, we build upon efficient REINFORCE-style algorithms. Through theoretical analysis and empirical validation, we identify ReMax (Li et al., 2024) as a superior backbone compared to GRPO (Shao et al., 2024) for regression tasks, as it avoids the detrimental scale invariance introduced by reward normalization. Finally, we validate GenRe$^2$ across three distinct domains: tabular regression on the TALENT benchmark (Ye et al., 2024a; Liu et al., 2025a), code metric regression (Akhauri et al., 2026) using RLM (Song et al., 2024; Akhauri et al., 2025; 2026), and Generative Reward Modeling (GRM) (Liu et al., 2025c; Mahdavi et al., 2025; Shi & Jin, 2025; Zhang et al., 2026b).

Our main contributions and findings are summarized as follows: (1) We propose GenRe$^2$ for decoding-based regression, a hybrid framework that integrates sequence-level RL with OPD; (2) We identify ReMax (Li et al., 2024) as superior to GRPO (Shao et al., 2024) for regression, proving that standard reward normalization causes detrimental scale invariance; (3) We reveal that OPD prevents RL-induced mode collapse, preserving entropy to enhance both sampling efficiency and prediction precision; (4) We validate the superiority of GenRe$^2$ across tabular regression, code metric regression, and GRM. Our code is available at https://github.com/lamda-bbo/GenRe2.

## 2. Background

Traditional DL-based regressors typically employ a point-wise head (predicting a scalar) or a Riemann head (predicting binned histogram distribution) (Imani & White, 2018), where the Riemann head has better robustness and performance in many applications and is widely used (Farebrother et al., 2024; Hollmann et al., 2025). A detailed overview of these methods is provided in Appendix B.1. Recently, Song & Bahri (2025) proposed decoding-based regression by reformulating regression as a discrete sequence generation

task, calling for a paradigm shift to generative regression. Specifically, a target scalar value $y$ is transformed into a sequence of discrete tokens $\mathcal{T} = \{t_1, t_2, \cdots, t_K\}$. Then, an autoregressive decoder head is trained to predict the tokens sequentially. Given the input representation $\phi(\mathbf{x})$, it models the conditional probability distribution $p_{\boldsymbol{\theta}}(y|\mathbf{x})$ as

$$p_{\boldsymbol{\theta}}(y|\mathbf{x}) = \prod_{k=1}^{K} p_{\boldsymbol{\theta}}(t_k \mid \phi(\mathbf{x}), \mathcal{T}_{<k}),$$

where $\mathcal{T}_{<k}$ denotes the tokens generated before step $k$. Given a dataset $\mathcal{D} = \{(\mathbf{x}_i, y_i)\}_{i=1}^{N}$, we first tokenize each target $y$ into a corresponding token sequence $\mathcal{T}$, then the decoder head is trained to predict the next token by minimizing the standard Cross-Entropy (CE) loss:

$$\mathcal{L}(\boldsymbol{\theta}) = -\mathbb{E}_{(\mathbf{x}, \mathcal{T}) \sim \mathcal{D}} \left[ \sum_{k=1}^{K} \log p_{\boldsymbol{\theta}}(t_k \mid \phi(\mathbf{x}), \mathcal{T}_{<k}) \right].$$

For inference, we generate $m$ candidate solutions via sampling (e.g., temperature sampling) and return the aggregation of these solutions. Here, the aggregation strategy can be various, such as $\mathrm{mean}(\cdot)$ or $\mathrm{median}(\cdot)$, and different aggregations lead to Bayes-optimal solutions for different regression metrics (Lukasik et al., 2024).

The tokenization of decoding-based regression is important. Following (Song & Bahri, 2025), we briefly introduce two common tokenization strategies (Detailed description can be founded in Appendix C):

- *Normalized Tokenization*: The normalized tokenization first scales a target value $y$ to a fixed interval (e.g. $[0, 1]$), then represents the scaled value as a base-$B$ expansion (e.g., 0.5 as <1><0><0> with $B = 2$). While effective, it relies on the access to the global minimum and maximum and is highly sensitive to outliers (Yeo & Johnson, 2000; Song et al., 2024).

- *Scientific Notation Tokenization*: Scientific notation tokenization methods (e.g., P10 (Charton, 2022) or IEEE (IEEE, 2019)) do not normalize the target, representing numbers using sign, mantissa, and exponent components (e.g., P10 represents 1.23 as <+><1><2><3><E-2>). They support a wider range of values but can be prone to yield hallucinations in unbounded generation (Song et al., 2024).

Intuitively, decoding-based regression generalizes histogram-based regression (e.g., Riemann head) into a multi-step binning paradigm, where tokenization defines the structure and the autoregressive decoding sequentially refines predictions (Song & Bahri, 2025). This approach offers two clear advantages: (1) It integrates seamlessly with LLMs, thereby enabling universal regression (Song et al., 2024) on free-formed inputs (Akhauri et al., 2025) while leveraging rich priors (Akhauri et al., 2026; Dong

et al., 2025); (2) It improves calibration. As noted in the reward model community (Mahan et al., 2024), sequential generative scoring yields more robust predictions and better mitigates reward hacking compared to scalar or histogram baselines (Yu et al., 2025).

Decoding-based regression has been applied to many downstream scenarios (Nguyen et al., 2025; You et al., 2026; Kim et al., 2025), one representative of which is RLM (Song et al., 2024). RLM directly regresses in the form of natural language, eliminating the need of feature engineering, which has been successfully applied to industrial scenarios (Akhauri et al., 2025), code metric prediction (Akhauri et al., 2026), and black-box optimization (Nguyen et al., 2024; Tan et al., 2025a; Zhang et al., 2025g; 2026a).

## 3. Method

In this section, we present our proposed method, GenRe$^2$, which leverages RL to address the sequence-level challenge of decoding-based regression with policy gradient. In Section 3.1, we first discuss the limitations of previous token-level decoding-based regression methods, showing emergent need for sequence-level supervisions and motivating us to solve it via RL. In Section 3.2, we formulate the decoding-based regression task as an MDP (Puterman, 2014), which serves as the foundation of GenRe$^2$. In Section 3.3, we formulate the regression reward. In Section 3.4, we theoretically justify our choice of ReMax over GRPO, identifying the detrimental *scale invariance* induced by standard normalization. In Section 3.5, we address the *temporal credit assignment* problem by augmenting RL with on-policy distillation (OPD). Finally, in Section 3.6, we analyze the gradient dynamics of various standard regularizers, theoretically demonstrating how GenRe$^2$ prevents mode collapse through targeted entropy-increasing tokens.

### 3.1. Limitations of Previous Token-level Methods

Standard decoding-based regression typically relies on CE. However, the potential of CE-trained decoding-based regression remains locked. Lukasik et al. (2025); Selvam (2025) theoretically showed that CE is not well-aligned with regression, as it treats digits as individual categories and ignores the numerical continuity. While recent improvements like NTL (Zausinger et al., 2025) and DIST$^2$ (Chung et al., 2026) introduce distance penalties, they still operate locally on individual tokens. As illustrated in the left part of Figure 2, the token-level losses overlook the global magnitude of the detokenized number. For instance, consider a target $y = 500$. At the first step, a low probability assignment implies a potential error of 100 (e.g., predicting '6'); at the last step, the same low probability implies a minor error of 1 (e.g., predicting '1'). Yet, if the model assigns identical weights to the correct tokens, CE yields identical losses.

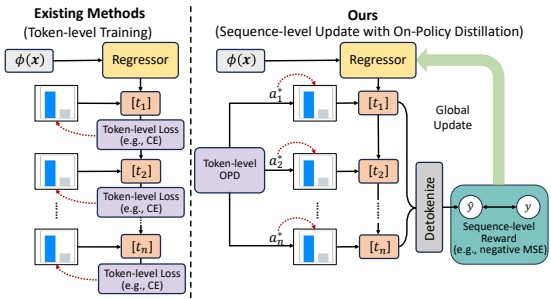

*Figure 2.* Comparison between local token-level training and global sequence-level update. **Left** (existing methods): The model is trained at each token $[t_1, \ldots, t_n]$ with a local loss (e.g., CE) that focuses solely on individual tokens. **Right** (ours): The model generates a full sequence $[t_1, \ldots, t_n]$ which is detokenized into a prediction $\hat{y}$. The optimization combines two signals: Global Update driven by a sequence-level reward, and token-level OPD which applies an on-policy distillation loss by comparing the sampled tokens against the optimal next tokens $a^*$ at each step.

This token-level limitation prevents CE from perceiving the global numerical error, causing the surrogate loss to deviate from the true regression objective.

### 3.2. Sequence-Level Optimization via RL

To bridge this gap, we reformulate the task via RL, optimizing the decoder using policy gradients with full-sequence rewards. This approach is motivated by recent successes in RL for LLM (Li et al., 2025b), where policy gradient methods (Sutton, 1988; Sutton et al., 1999) effectively align LLMs' responses with sequence-level, non-differentiable objectives, such as human preference (Christiano et al., 2017; Ouyang et al., 2022) or verifiable correctness (Guo et al., 2025; Wang et al., 2026a). We provide a detailed overview of RL for LLMs in Appendix B.2. We align with the standard autoregressive paradigm (formal MDP details in Appendix D): the state consists of the input features and the generated token history, the action corresponds to the next-token selection from the vocabulary, and the transition is the deterministic appending of the selected token. The reward is sparse, which is provided only after the complete numerical sequence is generated. Given the deterministic state transitions in the MDP, simple policy gradient methods like REINFORCE (Williams, 1992) are efficient. To determine the optimization backbone, we investigate two prevalent REINFORCE-style algorithms: ReMax (Li et al., 2024) and GRPO (Shao et al., 2024) (details provided in Appendix E). In the following two sections, we first define the precise regression reward, followed by a detailed analysis of the optimization algorithms that motivates our adoption of ReMax over GRPO.

### 3.3. Reward Formulation

The reward function is designed to guide the model towards the final regression metrics. Upon completing an episode,

the generated sequence $\tau$ is detokenized into its original prediction $\hat{y} = \text{Detokenize}(\tau)$. Given a bijective mapping $\psi$, we can define the terminal reward via the distance-based metrics in the target space, e.g., via the negative Mean Squared Error (MSE):

$$R(\tau) = -(\psi(\hat{y}) - \psi(y))^2,$$

where $y$ is the ground-truth target [1]. The mapping $\psi$ can be chosen flexibly, e.g., identity or normalization. This reward function is calculated on sequence level, and thus inherently numerically aware on the target space, which is a property being ignored in previous token-level objectives (Zausinger et al., 2025; Chung et al., 2026). It correctly assigns a relative higher reward to a numerically close prediction (e.g., 101 for a target of 100) compared to a numerically distant one (e.g., 200), even if both differ by a single token. This directly forces the model to learn the principles of numerical magnitude and proximity. We will discuss different settings of $\psi$ according to different problem natures in Section 4.

Notably, compared to RL with Verifiable Reward (RLVR) in LLM (Li et al., 2025b; Zhang et al., 2025c), which takes sparse reward, e.g., $\{-1, +1\}$, the reward in GenRe[2] is a dense one, where different generated sequences receive different rewards. With the sequence-level reward formally defined, the remaining challenge lies in selecting a policy gradient algorithm capable of effectively maximizing this objective. In the next section, we evaluate prevalent optimization backbones to identify the mechanism best suited for regression tasks.

### 3.4. RL Backbone: ReMax instead of GRPO

To determine the optimal backbone, we conduct a comparative study on tabular regression tasks (experimental details in Section 4.1). Our investigation proceeds from empirical observation to theoretical verification: we first identify the performance gap, then analytically trace its roots, and finally corroborate the mechanism through gradient visualization.

**Performance Gap and Component Ablation.** The left sub-figure in Figure 3 presents the performance comparison alongside a component ablation. We observe that ReMax consistently outperforms standard GRPO, and simply removing the normalization term (denoted as GRPO w/o std) restores the performance to a level comparable to ReMax. This finding effectively isolates the standard deviation (std) scaling in the advantage function as the primary source of the degradation.

**Theoretical Analysis: Gradient Scale Alignment.** To understand why the normalization term std is detrimental, we

---

[1]Given the inherent quantization error by discrete tokenization (Selvam, 2025), we round the target $y$ to the nearest tokenization bin to calculate the metrics.

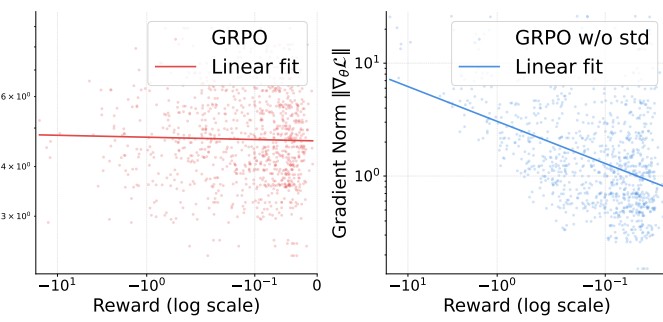

*Figure 3.* Comparison of optimization performance and stability. **(Left)** Ablation study showing ReMax and GRPO w/o std (removing normalization) outperform standard GRPO. Other variants include w/ Greedy Base. (using greedy rollout as baseline) and w/o IS.Clip (removing importance sampling clipping). **(Middle & Right)** Gradient norm vs. reward. Unlike standard GRPO (Middle), where normalization causes large gradients even for high-reward samples, GRPO w/o std (Right) correctly exhibits vanishing gradients for high-reward samples, ensuring stable convergence.

analyze the mathematical properties of the gradient magnitudes. Let $y$ denote the ground-truth target and $\hat{y}$ be the prediction detokenized from the generated sequence $\tau$. We define the absolute error as $E = |\hat{y} - y|$. In standard regression with the MSE loss, the gradient is $\nabla \mathcal{L}_{\text{MSE}} \propto (\hat{y} - y) \cdot \nabla \hat{y}$, where the coefficient $(\hat{y} - y)$ scales linearly with the absolute error $E$, implying that samples with less accurate predictions naturally yield larger update gradients.

We examine whether RL algorithms preserve this vital *scale-sensitive* property. For ReMax, the advantage is defined as $A_{\text{ReMax}} = R(\tau) - R(\tau_{\text{greedy}})$. Assuming a negative MSE reward $R(\hat{y}) = -(\hat{y} - y)^2$, we can expand the ReMax advantage using the difference of squares:

$$A_{\text{ReMax}} = -(\hat{y} - y)^2 - \left[-(\hat{y}_{\text{greedy}} - y)^2\right]$$
$$= (\hat{y}_{\text{greedy}} - y)^2 - (\hat{y} - y)^2$$
$$= \underbrace{(E_{\text{greedy}} - E_{\text{sample}})}_{\text{Relative Improvement}} \cdot \underbrace{(E_{\text{greedy}} + E_{\text{sample}})}_{\text{Absolute Scale Term}},$$

where $E_{\text{sample}}$ and $E_{\text{greedy}}$ denote the error magnitudes of the sampled and greedy rollouts, respectively. Notably, the second term acts as a scale-aware weight. Similar to MSE, ReMax naturally assigns larger gradient magnitudes to samples with higher absolute errors, prioritizing the correction of severe deviations.

In contrast, GRPO computes the advantage via normalization: $A_{\text{GRPO}} = (R - \mu)/\sigma$. The division by $\sigma$ effectively eliminates the absolute scale of the rewards. As a result, a batch with large errors yields normalized advantages in the same range as a batch with small errors. This *scale invariance* disconnects the gradient magnitude from the regression error, preventing the model from prioritizing the correction of severe mistakes.

**Empirical Verification: Gradient Dynamics.** To verify if this theoretical misalignment manifests in training, we visualize the gradient magnitudes for high reward samples

in Figure 3 (Middle & Right), where smaller gradients are preferred. For a direct comparison, we contrast standard GRPO against GRPO w/o std. The visualization corroborates our analysis: GRPO w/o std exhibits a clear correlation where high-reward samples (small errors) yield vanishing gradients, allowing the model to settle. In contrast, standard GRPO shows a disordered distribution where even high-reward samples generate large gradients due to the scaling by std. This confirms that normalization distorts the optimization landscape, preventing the model from converging on "solved" instances. Notably, while Dr. GRPO (Liu et al., 2025b) similarly removes std to prevent advantage explosion from near-zero variance in binary tasks, our analysis highlights a distinct failure mode for regression: standard normalization erases absolute-scale information, thereby destroying error-severity discrimination. Based on the theoretical alignment with the regression objective and the empirical superiority observed, we select ReMax as the RL backbone for GenRe².

### 3.5. GenRe²: Augmenting RL with On-Policy Distillation for Better Credit Assignment

**Challenge of Sparse Rewards.** While ReMax effectively aligns the gradient magnitude with the global error scale, it remains a sparse reward algorithm: reward is provided only after the entire sequence is generated. This creates a severe *temporal credit assignment problem*. Unlike natural language, where errors are often ambiguous, numerical regression is strictly hierarchical: an error in the Most Significant Token (MST) is catastrophic. However, while the final reward magnitude can indirectly reflect error severity, it fails to inform the model which specific token caused the error. Consequently, the policy cannot trace the source of a low reward, failing to identify whether it stems from a fundamental structural mistake (wrong magnitude) or a minor precision error.

**Empirical Pitfalls of ReMax.** To investigate the limitations of pure sequence-level RL in regression, we monitor both the accuracy of the MST and the policy entropy during training (detailed experimental settings are provided in Appendix F.2). Figure 4 illustrates two critical pitfalls. First, the left sub-figure shows that vanilla ReMax exhibits degradation on the MST, indicating its inability to reliably capture high-order magnitudes under sparse rewards. Second, the right sub-figure highlights a severe mode collapse, where the policy entropy of ReMax plummets. Together, these findings demonstrate that relying solely on sequence-level optimization is insufficient to maintain structural accuracy and necessary exploration.

**GenRe² Algorithm.** Building on these empirical pitfalls, and motivated by (Li et al., 2025a) which combines RL with imitation supervision, we propose GenRe² to integrate token-level supervision into the RL framework by directly employing OPD (Agarwal et al., 2024). To construct the teacher for OPD in the regression context, we define an expert oracle $\pi^*$ based on the ground-truth target $y$. For each step $t$ within a generated trajectory $\hat{y}_{<t}$, the expert determines the optimal next token $a_t^*$ that minimizes the distance to $y$. For instance, consider a target $y = 52$. If the model incorrectly samples 6 at the first step, the expert recognizes that the prefix is already overestimated. Consequently, it identifies 0 as the optimal next token to mitigate the error (forming 60, which is closer to 52 than 62), rather than blindly forcing the original target digit. We then augment the ReMax objective with this distillation loss:

$$\mathcal{L}(\theta) = \mathcal{L}_{\text{ReMax}}(\theta) + \lambda \cdot \mathbb{E}_{\tau \sim \pi_\theta} \left[ \sum_t -\log \pi_\theta(a_t^* | \hat{y}_{<t}) \right].$$

By providing immediate token-level feedback on active roll-outs, this OPD mechanism effectively mitigates the temporal credit assignment problem. Revisiting Figure 4, GenRe² yields consistent improvement on the MST accuracy, validating that token-level distillation from the expert oracle is essential for correctly learning hierarchical numerical structures. Meanwhile, GenRe² maintains a consistently higher entropy compared to ReMax, confirming that the OPD mechanism acts as an effective regularizer.

### 3.6. Theoretical Advantages over Standard RL Stabilizers

Incorporating auxiliary regularization terms is a prevalent practice to mitigate mode collapse. Common strategies include explicit entropy bonuses, Kullback-Leibler (KL) penalties, and standard supervised CE (Ouyang et al., 2022). However, these standard stabilizers exhibit fundamental limitations in continuous numerical regression. As illustrated by the $R^2$ dynamics in the left sub-figure of Figure 5, GenRe² consistently outperforms these general purposed

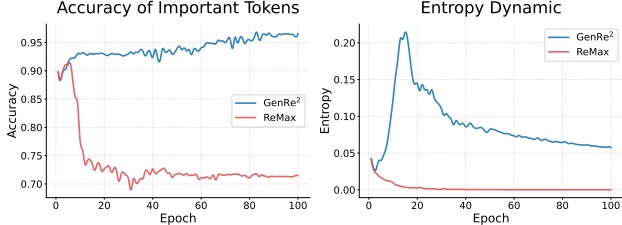

*Figure 4.* Learning dynamics of ReMax and GenRe². **(Left)** GenRe² consistently improves validation accuracy on important tokens, whereas ReMax degrades after early epochs. **(Right)** Unlike the premature entropy collapse observed in ReMax, GenRe² maintains exploration during training.

methods. To understand the underlying mechanism behind this performance, we analyze the gradient dynamics of various regularization terms, building upon the theoretical framework established by Wang et al. (2026b).

Let $z_k$ denote the pre-softmax logit for a candidate token $k$, and let $\pi_k = \frac{\exp(z_k)}{\sum_j \exp(z_j)}$ represent its corresponding policy probability. The system entropy is defined as $H = -\sum_j \pi_j \log \pi_j$.

**Proposition 3.1** (Entropy-Increasing Condition). *The gradient of the system entropy $H$ with respect to a specific logit $z_k$ is given by*

$$\frac{\partial H}{\partial z_k} = \pi_k(-\log \pi_k - H)$$

*Consequently, increasing a token's probability strictly raises the overall system entropy if and only if its current probability satisfies the condition:*

$$\pi_k < \exp(-H)$$

Based on Proposition 3.1 (the detailed proof is provided in Appendix A), we critically examine the limitations of standard regularization techniques in numerical regression and highlight the theoretical superiority of GenRe²:

**Inefficiency of Explicit Entropy Bonuses.** An explicit entropy bonus $+\alpha H$ forces a positive logit update for any token $k$ satisfying $\pi_k < \exp(-H)$. In numerical generation, numerous incorrect and catastrophic digit tokens also satisfy this low-probability condition. Consequently, the entropy bonus disperses probability mass non-selectively. Although it promotes generative diversity, it lacks clear directional guidance toward the numerical optimum.

**Ineffective Anchoring of KL Penalties.** Minimizing the KL divergence against a reference model anchors the policy to that prior. If the base model lacks numerical precision, the KL penalty preserves this uncertainty rather than sharpening the distribution toward the correct numerical neighborhood.

**Failure of Standard CE.** Standard supervised CE strictly enforces teacher-forcing based on the ground-truth (GT)

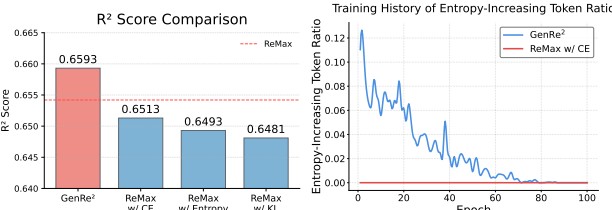

*Figure 5.* Ablation on stabilizers and training dynamics. **(Left)** scores of different stabilizers against the ReMax baseline. Only GenRe$^2$ yields a performance improvement. **(Right)** Ratio of entropy-increasing tokens over epochs. GenRe$^2$ successfully targets entropy-increasing tokens during early training, whereas ReMax w/ CE completely misses them.

tokens. From an entropy perspective, GT tokens typically possess a high prior probability, meaning they often fail to satisfy the entropy-increasing condition $\pi_k < \exp(-H)$. Consequently, maximizing GT token probabilities does little to inject entropy or prevent mode collapse.

**The Advantage of GenRe$^2$.** When generation deviates, the GenRe$^2$ oracle actively targets low-probability recovery tokens that minimize the numerical error. These tokens are likely to be low-probability candidates under the current policy, making them highly probable to satisfy the entropy-increasing condition $\pi_k < \exp(-H)$. This mechanism naturally injects entropy to prevent mode collapse while simultaneously optimizing regression precision. As the right sub-figure of Figure 5 confirms, GenRe$^2$ successfully isolates and targets these entropy-increasing tokens, enabling superior predictive performance.

# 4. Experiments

In this section, we empirically compare GenRe$^2$ with a variety of baseline methods on three representative decoding-based regression tasks. In Section 4.1, we evaluate GenRe$^2$ on tabular regression tasks, while in Section 4.2, we conduct experiments on the recently proposed Regression Language Model (RLM) to address code-to-metric regression. In Section 4.3, we extend our evaluation to Generative Reward Models (GRMs). Finally, we deliver some empirical discussions to understand the superior performance of RL in Section 4.4. Unless otherwise specified, we defer implementation details, hyperparameters, and evaluation protocols for all experiments to Appendix G. Note that we also implement our model equipped with ReMax (Li et al., 2024), which can show the general benefit of RL methods. The comparison between GenRe$^2$ and ReMax can also examine the effectiveness of using token-level supervision.

## 4.1. Tabular Regression

We examine the ability of GenRe$^2$ to perform tabular regression on the TALENT benchmark (Ye et al., 2024a; Liu et al., 2025a), a popular benchmark for tabular data containing 100

regression datasets. Following the practice of conducting RL after SFT in LLM post-training, we start RL from the CE-pretrained checkpoints. In Section 4.1.1, we introduce our experimental settings. Then we present the results to show the superiority of GenRe$^2$ in Section 4.1.2.

### 4.1.1. COMPARED METHODS

We mainly consider two categories of methods: (1) Baselines with different regression heads (i.e., pointwise head and Riemann head); (2) Decoding-based regression methods, including two NTL variants (NTL-WAS and NTL-MSE) (Zausinger et al., 2025) and DIST$^2$ (Chung et al., 2026). Here, NTL and DIST$^2$ are improvement methods for decoding-based regression with token-level loss. Details of the compared baselines can be found in Appendix G.1. Following (Song & Bahri, 2025), we instantiate the encoder $\phi$ as an MLP, and the decoder is a standard Transformer decoder.

### 4.1.2. RESULTS

In Table 1, we report the main results of our tabular regression experiments. We can observe that: (1) The base model of the decoder head method consistently outperforms the Pointwise baseline, validating the effectiveness of decoding based regression. (2) While token-level methods (NTL variants and DIST$^2$) yield marginal improvements over the base model, they fail to bridge the gap towards the robust Riemann baseline. (3) RL-based approaches, ReMax and GenRe$^2$, significantly improve the performance of the base model. Specifically, ReMax surpasses all token-level variants, while GenRe$^2$ further outperforms the robust Riemann baseline. These results imply that while sequence-level optimization is crucial for surpassing local supervision (as shown by ReMax), it is the integration of token-level expert signals in GenRe$^2$ that resolves the optimization bottleneck, enabling the model to finally exceed the strong Riemann baseline.

## 4.2. RLM for Code Metric Regression

In this section, we conduct experiments on the RLM (Song et al., 2024; Akhauri et al., 2025), an important downstream application of decoding-based regression, to perform code metric regression, following (Akhauri et al., 2026). Based on the input data format, we categorize the employed datasets into two streams:

- **High-level Source Code:** This category involves predicting runtime metrics directly from raw code snippets.

- **Computational Graph (ONNX):** This category focuses on Neural Architecture Search (NAS) benchmarks. The goal is to estimate the accuracy or latency of neural networks represented as serialized ONNX graphs.

*Table 1.* RMSE, $R^2$ and Spearman's Rank Correlation results over 5 random seeds on 100 TALENT regression tasks. The best and runner-up results are **bolded** and underlined, respectively. For the decoding-based methods, "Median" and "Mean" denote the aggregation strategy used to derive the final prediction from the generated candidates. The rows shaded in gray indicate our proposed methods.

| Head | Method | RMSE $\downarrow$ | | $R^2$ $\uparrow$ | | Rank Corr. $\uparrow$ | |
| --- | --- | --- | --- | --- | --- | --- | --- |
| | | Median | Mean | Median | Mean | Median | Mean |
| Pointwise | / | $0.5545_{\pm 0.0073}$ | $0.5545_{\pm 0.0073}$ | $0.5628_{\pm 0.0259}$ | $0.5628_{\pm 0.0259}$ | $0.7334_{\pm 0.0155}$ | $0.7334_{\pm 0.0155}$ |
| Riemann | / | $0.5152_{\pm 0.0025}$ | $0.5152_{\pm 0.0025}$ | $\underline{0.6547}_{\pm 0.0037}$ | $0.6547_{\pm 0.0037}$ | $0.7746_{\pm 0.0023}$ | $\underline{0.7746}_{\pm 0.0023}$ |
| Decoder | Base Model | $0.5293_{\pm 0.0008}$ | $0.5148_{\pm 0.0009}$ | $0.6223_{\pm 0.0009}$ | $0.6437_{\pm 0.0008}$ | $0.7682_{\pm 0.0033}$ | $0.7634_{\pm 0.0007}$ |
| | +CE | $0.5229_{\pm 0.0014}$ | $0.5074_{\pm 0.0009}$ | $0.6256_{\pm 0.0018}$ | $0.6498_{\pm 0.0009}$ | $0.7720_{\pm 0.0038}$ | $0.7672_{\pm 0.0011}$ |
| | +NTL-WAS (Zausinger et al., 2025) | $0.5378_{\pm 0.0008}$ | $0.5208_{\pm 0.0012}$ | $0.6235_{\pm 0.0011}$ | $0.6496_{\pm 0.0008}$ | $0.7662_{\pm 0.0024}$ | $0.7700_{\pm 0.0012}$ |
| | +NTL-MSE (Zausinger et al., 2025) | $0.5356_{\pm 0.0007}$ | $0.5206_{\pm 0.0009}$ | $0.6260_{\pm 0.0009}$ | $0.6496_{\pm 0.0012}$ | $0.7701_{\pm 0.0044}$ | $0.7696_{\pm 0.0012}$ |
| | +DIST$^2$ (Chung et al., 2026) | $0.5392_{\pm 0.0022}$ | $0.5608_{\pm 0.0018}$ | $0.6180_{\pm 0.0065}$ | $0.5101_{\pm 0.0069}$ | $0.7659_{\pm 0.0019}$ | $0.7384_{\pm 0.0014}$ |
| | **+ReMax** | $\underline{0.5035}_{\pm 0.0012}$ | $\underline{0.4994}_{\pm 0.0016}$ | $0.6499_{\pm 0.0015}$ | $\underline{0.6549}_{\pm 0.0022}$ | $\mathbf{0.7831}_{\pm 0.0045}$ | $0.7741_{\pm 0.0006}$ |
| | **+GenRe$^2$ (Ours)** | $\mathbf{0.4930}_{\pm 0.0018}$ | $\mathbf{0.4903}_{\pm 0.0013}$ | $\mathbf{0.6587}_{\pm 0.0016}$ | $\mathbf{0.6624}_{\pm 0.0015}$ | $\underline{0.7826}_{\pm 0.0038}$ | $\mathbf{0.7786}_{\pm 0.0011}$ |

We provide the specific statistics, regression targets, and detailed descriptions of these datasets in Appendix I.

### 4.2.1. COMPARED METHODS

We consider several decoding-based baselines for finetuning the given RLM checkpoint [2], including CE, NTL-WAS, NTL-MSE (Zausinger et al., 2025), and DIST$^2$ (Chung et al., 2026).

### 4.2.2. RESULTS

Table 2 summaries the results of different regression metrics on the NAS tasks, APPS Leetcode (Hendrycks et al., 2021), and Triton Kernel Latency (Paliskara & Saroufim, 2025) datasets. GenRe$^2$ demonstrates exceptional efficacy, securing the top performance across the majority of datasets. Notably, GenRe$^2$ overcomes the performance degradation seen in token-level methods on APPS and Triton, demonstrating its robustness. The sequence-level advantage is further validated by the consistent dominance of our RL approaches (i.e., ReMax and GenRe$^2$) across NAS tasks. The superiority of GenRe$^2$ over ReMax in all cases except NASNet confirms the benefit of integrating OPD signals. Detailed results on another high-level source code dataset CodeNet are provided in Appendix H.1.

### 4.3. Generative Reward Models

Next, we extend our evaluation to GRMs (Zhang et al., 2026b; Mahdavi et al., 2025; Shi & Jin, 2025), a pivotal component in the modern LLM post-training paradigm. Following the experimental setting of TRACT (Chiang et al., 2025), we finetune our model on the *Feedback Collection* dataset (Kim et al., 2024a). To evaluate the generalization capability of the learned reward model, we test it on a diverse suite of benchmarks. Detailed descriptions of all training and test datasets are provided in Appendix J.

---

[2] https://huggingface.co/akhauriyash/RLM-GemmaS-Code-v0

### 4.3.1. COMPARED METHODS

We compare our RL methods against token-level baselines. TRACT (Chiang et al., 2025) and prometheus-8x7b-v2.0 (Kim et al., 2024b) are excluded from direct baselines due to training data and parameter scale differences. They are both adapted to serve as teachers for OPD. To handle open-ended CoT generation, we substitute cross-entropy with KL divergence, which is similar to (Li et al., 2025a; Zhao et al., 2026).

### 4.3.2. RESULTS

As shown in Table 3, GenRe$^2$ consistently outperforms token-level and ReMax baselines using either OPD teacher. Notably, GenRe$^2$-prometheus (with a 7B backbone) achieves the highest average score of 0.6085, remarkably surpassing its 56B teacher (0.6060). This confirms that OPD effectively stabilizes CoT generation and distills high-quality reasoning capabilities.

### 4.4. Understanding the Effectiveness of GenRe$^2$ for Decoding-based Regression

Finally, we conduct illustrative experiments on tabular regression tasks to understand the effectiveness of GenRe$^2$ in decoding-based regression. In the context of RLVR, Yue et al. (2025) observed a critical phenomenon: while RL significantly boosts sampling efficiency (improving pass@1), it often fails to exceed the base model's theoretical potential, frequently underperforming on pass@$k$ at large $k$. This suggests that RL primarily sharpens the output distribution rather than expanding reasoning capacity.

However, standard regression metrics (e.g., MSE) merge all outcomes, failing to show the model's best possible output. To distinguish peak capacity from average sampling, we introduce a **best@$k$** metric under oracle selection. Specifically, for a given feature $\phi(\mathbf{x}_i)$ in the test set, the model generates $k$ predictions $\{\hat{y}_i^1, \ldots, \hat{y}_i^k\}$ and selects the closest

*Table 2.* Results (i.e., Spearman's Rank Correlation) for code metric regression on NAS datasets, APPS Leetcode and Triton Kernel Latency. We train models and report results as the average over 5 random seeds. The best and runner-up results are **bolded** and underlined, respectively. The rows shaded in gray indicate our proposed methods.

| Model | NASBench101 | NASBench201 | Amoeba | DARTS | ENAS | NASNet | PNAS | APPS Leetcode | Triton Kernel Latency |
|---|---|---|---|---|---|---|---|---|---|
| Base Model | $0.680_{\pm 0.005}$ | $0.662_{\pm 0.003}$ | $0.128_{\pm 0.011}$ | $0.402_{\pm 0.010}$ | $0.265_{\pm 0.018}$ | $0.178_{\pm 0.019}$ | $0.267_{\pm 0.017}$ | $0.935_{\pm 0.000}$ | $0.536_{\pm 0.003}$ |
| +CE | $0.727_{\pm 0.006}$ | $0.658_{\pm 0.002}$ | $0.576_{\pm 0.033}$ | $0.728_{\pm 0.002}$ | $0.652_{\pm 0.017}$ | $0.434_{\pm 0.111}$ | $0.669_{\pm 0.002}$ | $0.913_{\pm 0.001}$ | $0.555_{\pm 0.001}$ |
| +NTL-WAS (Zausinger et al., 2025) | $0.711_{\pm 0.021}$ | $0.695_{\pm 0.012}$ | $0.464_{\pm 0.242}$ | $0.728_{\pm 0.003}$ | $\overline{0.665}_{\pm 0.008}$ | $0.460_{\pm 0.056}$ | $\underline{0.671}_{\pm 0.003}$ | $0.904_{\pm 0.001}$ | $0.539_{\pm 0.010}$ |
| +NTL-MSE (Zausinger et al., 2025) | $0.714_{\pm 0.014}$ | $0.673_{\pm 0.021}$ | $0.572_{\pm 0.018}$ | $0.689_{\pm 0.052}$ | $0.633_{\pm 0.008}$ | $0.372_{\pm 0.098}$ | $0.660_{\pm 0.005}$ | $0.867_{\pm 0.002}$ | $0.510_{\pm 0.008}$ |
| +DIST² (Chung et al., 2026) | $0.728_{\pm 0.004}$ | $0.687_{\pm 0.014}$ | $0.598_{\pm 0.004}$ | $0.729_{\pm 0.002}$ | $0.648_{\pm 0.009}$ | $0.504_{\pm 0.029}$ | $0.670_{\pm 0.002}$ | $0.902_{\pm 0.002}$ | $0.540_{\pm 0.006}$ |
| **+ReMax** | $\underline{0.772}_{\pm 0.006}$ | $\underline{0.713}_{\pm 0.009}$ | $\underline{0.604}_{\pm 0.012}$ | $\underline{0.737}_{\pm 0.006}$ | $0.624_{\pm 0.010}$ | $\mathbf{0.553}_{\pm 0.007}$ | $0.670_{\pm 0.011}$ | $\underline{0.970}_{\pm 0.000}$ | $0.618_{\pm 0.011}$ |
| **+GenRe² (Ours)** | $\mathbf{0.776}_{\pm 0.010}$ | $\mathbf{0.726}_{\pm 0.004}$ | $\mathbf{0.621}_{\pm 0.005}$ | $\mathbf{0.745}_{\pm 0.004}$ | $0.630_{\pm 0.016}$ | $\underline{0.539}_{\pm 0.003}$ | $\mathbf{0.679}_{\pm 0.006}$ | $\mathbf{0.977}_{\pm 0.000}$ | $\mathbf{0.631}_{\pm 0.013}$ |

*Table 3.* Results (i.e., Spearman's Rank Correlation) for Generative Reward Models on Reward benchmarks. We train models and report results as the average over 5 random seeds. The best and runner-up results are **bolded** and underlined, respectively. The rows shaded in gray indicate our proposed methods.

| Model | Param. | FB Bench | FLASK | MT Bench | Vicuna Bench | Avg. Score |
|---|---|---|---|---|---|---|
| Base Model | 7B | 0.5718 | 0.3944 | 0.2826 | 0.3477 | 0.3991 |
| +CE | 7B | 0.7913 | 0.3726 | 0.2908 | 0.3429 | 0.4494 |
| +DIST² | 7B | 0.6714 | 0.2403 | 0.3693 | 0.2440 | 0.3813 |
| **+ReMax** | 7B | 0.8742 | 0.3954 | 0.3346 | 0.4157 | 0.5050 |
| TRACT | 7B | **0.9090** | 0.4593 | 0.5105 | 0.5188 | 0.5994 |
| prometheus-8x7b-v2.0 | 56B | 0.7559 | **0.6160** | 0.5101 | 0.5418 | 0.6060 |
| **+GenRe²-TRACT** | 7B | 0.8984 | 0.4036 | 0.3785 | 0.3739 | 0.5136 |
| **+GenRe²-prometheus** | 7B | 0.8667 | 0.4773 | **0.5256** | **0.5644** | **0.6085** |

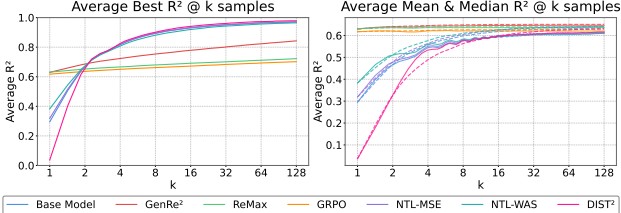

*Figure 6.* Metric dynamics across 100 TALENT regression tasks. The left sub-figure displays the average best $R^2@k$, while the right one shows the average mean (dashed) and median (solid) $R^2@k$.

one to the ground truth $y_i$:

$$\hat{y}_i^{\text{best}} = \arg\min_{\hat{y} \in \{\hat{y}_i^1, \dots, \hat{y}_i^k\}} |y_i - \hat{y}|.$$

Then, the best@$k$ metrics can be calculated using the collection of all $\hat{y}_i^{\text{best}}$ values.

We plot the best $R^2@k$ with varying $k$ in the left sub-figure of Figure 6. Consistent with the observations in RLVR (Zhu et al., 2025b), we find that the RL-tuned models significantly improve best $R^2@1$, but are surpassed by the base model as $k$ increases. Note that GenRe² exhibits better scaling than ReMax (suffering from severe mode collapse), which indicates that on-policy distillation effectively preserves diversity. While all token-level finetuning methods maintain performance at large $k$, RL methods implicitly suppress exploration of the output space, thereby lowering the capability boundary. As a result, this lower variance allows GenRe² to generate better solutions in a single trial (i.e., better best@1), thus achieving better mean and median regression performance shown in the right sub-figure of Figure 6.

### 4.5. Additional Analysis and Ablation Studies

We provide extensive additional results throughout the Appendix to further examine the robustness, mechanisms, and cost of GenRe². First, we study whether the proposed sequence-level optimization remains effective beyond normalized tokenization. Appendix H.2 evaluates scientific-notation tokenizers, including P10 and IEEE, showing that GenRe² is not tied to a specific numeric representation. Appendix H.3 then compares different RL backbones and analyzes which components of policy-gradient optimization are responsible for stable regression improvements. We further investigate several robustness factors: Appendix H.4 studies catastrophic outlier suppression and verifies whether sequence-level RL can reduce severe numerical failures, Appendix H.5 varies the numerical base used in tokenization to test sensitivity to token granularity, and Appendix H.6 demonstrates that quantile normalization provides a more stable reward scale for code metric regression. To better understand how GenRe² changes the learned predictor, Appendix H.7 visualizes its distributional impact, while Appendix H.8 examines whether assigning non-uniform weights to different token positions can further improve numerical precision. Finally, we further analyze the OPD mechanism and training behavior: Appendix H.9 evaluates GenRe² under the task setting of the original NTL work, Appendix H.10 details how boundary tokens correct invalid or suboptimal prefixes under different tokenizers, Appendix H.11 benchmarks standard RL regularizers such as entropy bonuses and KL penalties, and Appendix H.12 reports the computational overhead of GenRe².

### 5. Conclusion

In this paper, we propose to unlock the potential of decoding-based regression via RL. Specifically, we challenge the practice of training the model via token-level loss, and propose GenRe² to address the limitations of prior methods by integrating sequence-level RL with on-policy distillation. Experimental results on tabular regression, code metric regression and generative reward modeling show the superiority and generalization of GenRe², establishing generative models as a competitive paradigm for general-purpose numerical prediction and broader applications such as offline black-box optimization (Xue et al., 2024; Tan et al., 2025b).

## Impact Statement

This paper presents work whose goal is to advance the field of decoding-based regression in general-purpose numerical prediction tasks. There are many potential societal consequences of our work, none which we feel must be specifically highlighted here.

## Acknowledgments

This work was supported by the National Science Foundation of China (624B2069, 625B1018, 62276124), the Fundamental and Interdisciplinary Disciplines Breakthrough Plan of the Ministry of Education of China (JYB2025XDXM118), and the "111 Center" (B26023). Ke Xue and Chao Qian are the corresponding authors.

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

# A. Proof of Proposition 3.1

*Proof.* Let $\pi_i = \exp(z_i)/\sum_j \exp(z_j)$ be the softmax probability. The derivative of $\pi_i$ with respect to the logit $z_k$ is given by:

$$\frac{\partial \pi_i}{\partial z_k} = \pi_k(\delta_{ik} - \pi_i)$$

where $\delta_{ik}$ is the Kronecker delta. The system entropy is $H = -\sum_i \pi_i \log \pi_i$. Using the chain rule, the gradient of $H$ with respect to $z_k$ is:

$$\begin{aligned}
\frac{\partial H}{\partial z_k} &= \sum_i \frac{\partial H}{\partial \pi_i} \cdot \frac{\partial \pi_i}{\partial z_k} \\
&= \sum_i -(\log \pi_i + 1) \cdot \pi_k(\delta_{ik} - \pi_i) \\
&= -\pi_k(\log \pi_k + 1) + \pi_k \sum_i \pi_i(\log \pi_i + 1)
\end{aligned}$$

Since $\sum_i \pi_i = 1$ and $\sum_i \pi_i \log \pi_i = -H$, the summation term simplifies to:

$$\sum_i \pi_i \log \pi_i + \sum_i \pi_i = -H + 1$$

Substituting this back into the gradient expression:

$$\begin{aligned}
\frac{\partial H}{\partial z_k} &= -\pi_k \log \pi_k - \pi_k + \pi_k(-H + 1) \\
&= \pi_k(-\log \pi_k - H)
\end{aligned}$$

This completes the derivation of the gradient. To satisfy the entropy-increasing condition $\frac{\partial H}{\partial z_k} > 0$, and noting that $\pi_k > 0$, we require:

$$\begin{aligned}
-\log \pi_k - H &> 0 \\
\log \pi_k &< -H \\
\pi_k &< \exp(-H)
\end{aligned}$$

The proof is thus complete. □

# B. Additional Backgrounds

### B.1. Regression

Given a training dataset $\mathcal{D}_{\text{train}} = \{(\mathbf{x}_i, y_i)\}_{i=1}^N$ sampled from an unknown ground-truth function $f : \mathcal{X} \to \mathbb{R}$, regression aims to learn a model from $\mathcal{D}_{\text{train}}$ that accurately predicts the output for unseen inputs. The quality of the learned model is evaluated on a hold-out test set, $\mathcal{D}_{\text{test}}$, by measuring its predictive ability with regression-based metrics, e.g., Root Mean Squared Error (RMSE).

Traditional regression methods involve statistical techniques like Gaussian Processes (Rasmussen & Williams, 2006) and tree-based methods (Chen & Guestrin, 2016; Zhou & Feng, 2019). Recent works have focused on deep learning (DL)-based methods (Gorishniy et al., 2021; Jiang et al., 2026), which train deep neural networks to leverage the power of representation learning for regression, demonstrating great superiority and scalability (Gorishniy et al., 2021; Ye et al., 2024a). Specifically, DL-based methods train neural networks to map the input $\mathbf{x}$ to a high-dimensional representation $\phi(\mathbf{x})$, and subsequently models the probability distribution of the target value $p_{\boldsymbol{\theta}}(y \mid \phi(\mathbf{x}))$ via a regression head, parameterized by $\boldsymbol{\theta}$.

There are several design philosophies for the regression head, including *pointwise head*, *parametric distribution head*, and *Riemann head*. The *pointwise head* maps $\phi(\mathbf{x})$ to a scalar prediction, which is the most commonly used regression head. However, the pointwise head fails to capture both the uncertainty (Lakshminarayanan et al., 2017) and the complex multimodality of the target distribution (Bishop, 1994). To address this, the *parametric distribution head* instead models

output as a predefined distribution (e.g., Gaussian) and predicts its parameters (e.g., the mean value and the standard variance) (Garnelo et al., 2018a;b; Nguyen & Grover, 2022). The *Riemann head*, also called *histogram head*, instead converts the regression problem into classification by discretizing the continuous output $y$ into finite bins (Imani & White, 2018). The learned model predicts the probability of each bin, from which the output value is derived using a weighted sum. Though sensible to hyperparameters (Tarasov et al., 2024), the *Riemann head* has been shown to improve the model's robustness and performance (Imani & White, 2018; Imani et al., 2024), with successful application to reinforcement learning (RL) (Bellemare et al., 2017; Farebrother et al., 2024) and tabular foundation models (Hollmann et al., 2025; Grinsztajn et al., 2025; Zhang et al., 2025f).

### B.2. RL for LLM

Reinforcement Learning (RL) has become a pivotal post-training technique for LLMs (Li et al., 2025b), popularized by RLHF for alignment (Christiano et al., 2017; Ouyang et al., 2022; OpenAI, 2023), and extended to domains with verifiable rewards like mathematical (Guo et al., 2025; Wang et al., 2026a) and scientific reasoning (Bai et al., 2025; Chen et al., 2025b). RL approaches are primarily categorized into offline preference optimization (Rafailov et al., 2023; Meng et al., 2024; Ethayarajh et al., 2024) and online policy gradient methods (Schulman et al., 2017; Williams, 1992). While early online methods relied on actor-critic algorithms like PPO (Schulman et al., 2017), recent works leverage the deterministic transitions of LLMs to adopt lightweight REINFORCE-based methods without a value model (Williams, 1992). Notably, ReMax (Li et al., 2024) and GRPO (Shao et al., 2024) reduce variance using greedy and multi-sample mean baselines, respectively, with the latter showing power in DeepSeek-R1 (Guo et al., 2025). Further variants enhance scalability and stability through reducing variance (Hu et al., 2025) and estimation biases (Ahmadian et al., 2024; Liu et al., 2025b), and employing regularization techniques such as entropy (Cui et al., 2025) and reward shaping (Chen et al., 2025c; Peng et al., 2025). Compared to Supervised Finetuning (SFT), RL demonstrates superior generalization (Chu et al., 2025; Matsutani et al., 2026; Zhu et al., 2025a) and mitigated forgetting (Shenfeld et al., 2026; Chen et al., 2025a).

## C. Description of Tokenizations

Here we provide detailed descriptions of the two common tokenization strategies introduced in the main text:

- *Normalized Tokenization*: The normalization tokenization first scales a target value $y$ to a fixed interval (e.g. $[0, 1]$). Then, this method represents the scaled value as a base-$B$ expansion. For instance, when choosing $B = 2$ and a mantissa length of $M = 3$, the scaled number $0.6$ is tokenized as `<1><1><0>`. For prediction, we need rescale the detokenized number to its original space. However, normalized tokenization relies on the access to the global minimum and maximum. One could set $y_{\min}$ and $y_{\max}$ in accordance with the training dataset, but this method is highly sensitive to outliers (Yeo & Johnson, 2000). Besides, it is unsuitable for multi-task regression, where different tasks may have different objectives, as globally linear scaling to $[0, 1]$ can cause precision loss (Song et al., 2024).

- *Scientific Notation Tokenization*: Unlike normalized approaches, scientific notation tokenization methods do not normalize the target value. Instead, they represent numbers using sign, mantissa, and exponent components. We describe two specific implementations below:

  - *P10 Tokenization* (Charton, 2022): P10 Tokenization is an unnormalized tokenization method that represents numbers in a format similar to scientific notation. It breaks down a scalar into three components: A sign token, a mantissa part with $M$ tokens, and an exponent token. For example, with a mantissa length of $M = 3$, the number $1.23$ is tokenized as `<+><1><2><3><E-2>`.

  - *IEEE Tokenization* (IEEE, 2019): IEEE Tokenization is another unnormalized tokenization scheme that directly represents a target value $y$ by generalizing the IEEE-754 floating-point standard into a base-$B$ format. It tokenizes a number into a sequence representing its sign, exponent and mantissa. For instance, with base $B = 10$, an exponent length of $E = 3$, and a mantissa length of $M = 4$, the number $10^{-12} \times 1.234$ is tokenized as `<+><−><0><1><2><1><2><3><4>`.

## D. Problem Formulation

In this section, we formalize the generation of a numerical sequence (i.e., the primary goal of decoding-based regression) as an MDP. Specifically, taking the generation of 6 (the sequence representation is `<1><1><0>`) as an example, for an input representation $\phi(\mathbf{x})$, the MDP $\mathcal{M} = (\mathcal{S}, \mathcal{A}, P, r, \rho, T)$ can be written by:

- **State** $\mathcal{S}$: A state $s_k \in \mathcal{S}$ is defined by the input feature and the generated token sequence, i.e., $s_k = (\phi(\mathbf{x}), \mathcal{T}_{<k})$, where $\mathcal{T}_{<k} = (t_0, \ldots, t_{k-1})$. For instance, an intermediate state at $k = 2$ is $s_2 = (\phi(\mathbf{x}), \texttt{<1><1>})$.

- **Action** $\mathcal{A}$: The action space $\mathcal{A}$ is defined over the token vocabulary $\mathcal{V}$, where an action $a_k$ is the selection of the next token $t_k \in \mathcal{V}$. For instance, given the state $s_2$, the model can sample the next token $\texttt{<0>}$ or $\texttt{<1>}$ to proceed towards completing the sequence.

- **Transition** $P$: The state transitions $P(s_{k+1} \mid s_k, a)$ are deterministic. Appending a selected token $a_k = t_k$ to the current state $s_k$ always leads to a unique next state $s_{k+1} = (s_k, a_k) = (\phi(\mathbf{x}), \{\mathcal{T}_{<k}, t_k\})$, which is also an important characteristic of RL formulation in LLM (Li et al., 2024; 2025b). Continuing the example from state $s_2$, if the model samples the action $\texttt{<0>}$, the state transitions to a sequence $\texttt{<1><1><0>}$ (decoding to 6); conversely, if the action $\texttt{<1>}$ is sampled, the state updates to $\texttt{<1><1><1>}$ (decoding to 7).

- **Reward** $r$: The reward function $r$ assigns reward values to state-action pairs. Since we have to access signals from the detokenized numerical value only after the entire sequence is generated, the reward function is defined by:

$$r(s_k, a_k) = \begin{cases} 0 & \text{if } k \neq K - 1 \\ r(\phi(\mathbf{x}), a_{0:K-1}) & \text{otherwise.} \end{cases}$$

Consistent with the formulation of RL in LLM (Li et al., 2025b; Zhang et al., 2025c), this reward design is sparse with zero rewards to all intermediate generation steps. The specific design of $r(\phi(\mathbf{x}), a_{0:K-1})$ can be flexible, which we will elaborate in Section 3.3.

- **Initial State Distribution** $\rho$: The distribution $\rho$ is deterministic, with the initial state $s_0$ corresponding to the input feature $\phi(\mathbf{x})$ and an empty sequence.

- **Horizon** $T$: $T$ is the maximum length of the generated sequence, i.e., $T = K$.

Within this framework, the learning objective of GenRe$^2$ is to maximize the expected return:

$$\mathcal{J}(\pi_{\boldsymbol{\theta}}) = \mathbb{E}_{(\mathbf{x},y)\sim\mathcal{D}_{\text{train}}} \mathbb{E}_{\tau\sim\pi_{\boldsymbol{\theta}}} \left[ \sum_{k=0}^{K-1} r(s_k, a_k) \right], \tag{1}$$

where $\pi_{\boldsymbol{\theta}}$ is the policy parameterized by $\boldsymbol{\theta}$, and $\tau = (s_0, a_0, \cdots, s_K)$ denotes a trajectory sampled from $\pi_{\boldsymbol{\theta}}$.

By formulating the decoding regression task into a policy optimization problem, we employ the policy gradient method (Sutton et al., 1999) to optimize $\pi_{\boldsymbol{\theta}}$ by ascending the gradient of the expected return:

$$\nabla_{\boldsymbol{\theta}} \mathcal{J}(\pi_{\boldsymbol{\theta}}) = \mathbb{E}_{(\mathbf{x},y)\sim\mathcal{D}_{\text{train}}} \mathbb{E}_{\tau\sim\pi_{\boldsymbol{\theta}}}$$
$$\left[ \sum_{k=0}^{K-1} \nabla_{\boldsymbol{\theta}} \log \pi_{\boldsymbol{\theta}}(a_k \mid s_k) A^{\pi_{\boldsymbol{\theta}}}(s_k, a_k) \right], \tag{2}$$

where $A^{\pi_{\boldsymbol{\theta}}}(s_t, a_t)$ is the advantage function estimating the relative value of action $a_k$ in state $s_k$. Given the deterministic state transitions in the MDP, simple policy gradient methods like REINFORCE (Williams, 1992) are efficient.

## E. Policy Gradient Methods

To optimize the objective in Eq. (1), we employ the policy gradient method (Sutton et al., 1999) to optimize $\pi_{\boldsymbol{\theta}}$ by ascending the gradient of the expected return:

$$\nabla_{\boldsymbol{\theta}} \mathcal{J}(\boldsymbol{\theta}) = \mathbb{E}_{(\mathbf{x},y)\sim\mathcal{D}_{\text{train}}} \mathbb{E}_{\tau\sim\pi_{\boldsymbol{\theta}}} \left[ \sum_{k=0}^{K-1} \nabla_{\boldsymbol{\theta}} \log \pi_{\boldsymbol{\theta}}(a_k \mid s_k) A^{\pi_{\boldsymbol{\theta}}}(s_k, a_k) \right], \tag{3}$$

where $A^{\pi_{\boldsymbol{\theta}}}(s_k, a_k)$ is the advantage function estimating the relative value of action $a_k$ in state $s_k$. As the expectation $\mathbb{E}_{\tau\sim\pi_{\boldsymbol{\theta}}}$ is intractable, one practical solution is to approximate it via Monte Carlo sampling.

Given the deterministic state transitions in the MDP, simple policy gradient methods like REINFORCE (Williams, 1992) are efficient. To reduce variance, REINFORCE subtracts a baseline in the advantage function:

$$A^{\pi_\theta}(s_k, a_k) = R(\tau) - b(\phi(\mathbf{x})),$$

where $\tau = (s_0, a_0, \cdots, s_K)$ denotes a trajectory sampled from $\pi_\theta$, $R(\tau) = \sum_{k=0}^{K-1} r(s_k, a_k)$ is the expected return of the trajectory $\tau$, and $b(\phi(\mathbf{x}))$ is the baseline value related to the input $\phi(\mathbf{x})$, which is to be designed. Crucially, this subtraction maintains an unbiased estimator (Williams, 1992), forming the foundation of online policy gradient methods (Sutton & Barto, 2018).

In this paper, we employ two prevalent algorithms, ReMax (Li et al., 2024) and GRPO (Shao et al., 2024), which can be viewed as REINFORCE variants with distinct advantage formulations. ReMax reduces variance efficiently by setting the baseline as the reward of a greedy decoding sequence:

$$A^{\pi_\theta}_{\text{ReMax}}(\tau) = R(\tau) - r(\phi(\mathbf{x}), \hat{a}_{0:K-1}), \text{ where } \hat{a}_k \in \arg\max \pi_\theta(\cdot \mid \phi(\mathbf{x}), \hat{a}_{0:k-1}).$$

GRPO, on the other hand, computes the advantage by normalizing rewards relative to a group of $G$ sampled trajectories $\{\tau^i\}_{i=1}^G$:

$$A^{\pi_\theta}_{\text{GRPO}}(\tau^i) = \frac{R(\tau^i) - \text{mean}_j\left\{R(\tau^j)\right\}}{\text{std}_j\left\{R(\tau^j)\right\} + \epsilon},$$

where $\epsilon$ is a small positive constant. Additionally, GRPO further stabilizes training by incorporating importance sampling and clipping mechanisms into its final objective, which is defined as:

$$\mathcal{J}(\boldsymbol{\theta}) = \mathbb{E}_{\tau \sim \pi_\theta}\left[\frac{1}{G}\sum_{i=1}^G \min\left\{\text{IS}(\boldsymbol{\theta})A^{\pi_\theta}_{\text{GRPO}}(\tau^i), \text{clip}(\text{IS}(\boldsymbol{\theta}), 1-\varepsilon, 1+\varepsilon)A^{\pi_\theta}_{\text{GRPO}}(\tau^i)\right\}\right],$$

where $\text{IS}(\boldsymbol{\theta}) = \frac{\pi_\theta(\tau^i|\phi(\mathbf{x}))}{\pi_{\theta_{\text{old}}}(\tau^i|\phi(\mathbf{x}))}$ denotes the importance sampling ratio between the current and reference policies, and $\varepsilon$ is a hyperparameter controlling the clipping range.

## F. Experimental Settings for Figures and Tables

### F.1. Experimental Settings for Figure 3

To generate the optimization comparison in Figure 3, we utilize the TALENT benchmark. For the $R^2$ score ablation (Left), the experimental setup mirrors the main tabular regression evaluation, with the exception that we omit the automated hyperparameter search and manually fix the tokenization settings to base $B = 10$ and precision $P = 3$. For the gradient dynamics visualizations (Middle and Right), we extract and plot the training trajectory from a single representative task within the TALENT benchmark to clearly illustrate the correlation between reward scales and gradient magnitudes.

### F.2. Experimental Settings for Figure 4

To generate the visualization in Figure 4, we select a representative task from the TALENT benchmark and adopt the IEEE tokenization format with base $B = 10$ and exponent length $E = 1$. We define the Most Significant Tokens (MST) as the first three generated tokens: the sign, the exponent sign, and the exponent digit. The accuracy presented in the left panel represents the exact-match rate, where all three of these tokens are correctly predicted. Concurrently, the right panel tracks the corresponding entropy dynamics of the policy during the training process.

### F.3. Experimental Settings for Figure 5

The experimental configuration for Figure 5 strictly follows the primary tabular regression setup detailed in our main evaluation. For the $R^2$ score comparison (Left), the reported performance is aggregated across all evaluated tasks within the benchmark. Conversely, the training history of the entropy-increasing token ratio (Right) is tracked and visualized using a single representative task to clearly illustrate the token-level dynamics throughout the training process.

# G. Experimental Settings

## G.1. Baseline Details

In this section, we provide detailed implementations of the baselines compared in our experiments. Consistent with the main text, we categorize these methods into two groups: (1) **Baselines with different regression heads**, specifically the Pointwise head and the Riemann head; and (2) **Decoding-based regression methods**, which incorporate token-level loss improvements including NTL variants (NTL-MSE, NTL-WAS) and DIST$^2$.

### G.1.1. POINTWISE HEAD

The Pointwise head represents the standard regression approach. It projects the latent representation $\phi(\mathbf{x})$ directly to a scalar prediction $\hat{y} \in \mathbb{R}$ via a linear layer. The model is optimized by minimizing the Mean Squared Error (MSE) loss between the predicted value and the ground truth:

$$\mathcal{L}_{\text{MSE}} = \frac{1}{N} \sum_{i=1}^{N} (y_i - \hat{y}_i)^2 \, ,$$

where $N$ denotes the number of samples in the batch, $y_i$ and $\hat{y}_i$ denote the true value and the predicted value, respectively.

### G.1.2. RIEMANN HEAD

Following (Müller et al., 2022; 2023), we implement the Riemann head by combining the infinite support architecture with the histogram loss objective (Imani & White, 2018). This approach models the regression target as a probability distribution rather than a single scalar, allowing for better handling of uncertainty and outliers.

**Infinite Support Architecture.** We partition the target space of $y$ into a central finite range and two infinite tails to handle potential outliers. The central range $[y_{\min}, y_{\max}]$ is divided into $K$ uniform bins, each with width $w = (y_{\max} - y_{\min})/K$. Additionally, we define a left tail region for $y < y_{\min}$ and a right tail region for $y \geq y_{\max}$. The neural network outputs a probability vector $\boldsymbol{o} = [o_0, \ldots, o_{K+1}]$, representing the probability mass assigned to each component. The full predictive Probability Density Function (PDF), denoted as $q_{\boldsymbol{\theta}}(y \mid \mathbf{x})$, is defined piecewise:

$$q_{\boldsymbol{\theta}}(y \mid \mathbf{x}) = \begin{cases} \underbrace{\frac{o_0}{\sigma_{\text{tail}}} \phi_{HN}\left(\frac{y_{\min} - y}{\sigma_{\text{tail}}}\right)}_{\text{Left Tail (Half-Normal)}} & \text{if } y < y_{\min} \\[2ex] \underbrace{\frac{o_k}{w}}_{\text{Central Bins (Uniform)}} & \text{if } y \in [y_{\min} + (k-1)w, y_{\min} + kw) \, , k \in \{1, 2, \ldots, K\} \\[2ex] \underbrace{\frac{o_{K+1}}{\sigma_{\text{tail}}} \phi_{HN}\left(\frac{y - y_{\max}}{\sigma_{\text{tail}}}\right)}_{\text{Right Tail (Half-Normal)}} & \text{if } y \geq y_{\max} \end{cases}$$

where $\phi_{HN}(\cdot)$ is the PDF of the standard Half-Normal distribution, and $\sigma_{\text{tail}}$ is a fixed scale parameter (set to 0.5) controlling the decay rate in the tail regions.

**Histogram Loss.** To train the model, we construct a smoothed target distribution. Given a ground truth scalar $y_{gt}$, we model the target distribution $p(y)$ as a Gaussian centered at $y_{gt}$ with standard deviation $\sigma = 0.75w$, truncated to the central range: $p(y) \propto \mathcal{N}(y; y_{gt}, \sigma^2) \cdot \mathbb{I}[y_{\min}, y_{\max}]$. We then discretize this continuous target by integrating $p(y)$ over each bin's interval to obtain the target probability mass $p_k = \int_{l_k}^{r_k} p(y)dy$, where $[l_k, r_k]$ denotes the interval of the $k$-th central bin. The model is optimized by minimizing the CE loss between the target mass vector $\mathbf{p}$ and the predicted mass vector $\boldsymbol{o}$:

$$\mathcal{L} = -\sum_{k=0}^{K+1} p_k \log(o_k).$$

**Inference.** During inference, we obtain the final scalar prediction $\hat{y}$ by calculating the expected value of the predicted

distribution $q_\theta(y|x)$. This is computed as the weighted sum of the centroids of all components:

$$\hat{y} = \mathbb{E}_{y \sim q_\theta}[y] = \sum_{k=0}^{K+1} o_k \cdot c_k,$$

where $c_k$ is the centroid of the $k$-th component. For central bins, $c_k$ is the midpoint of the interval; for the tail regions, $c_k$ is the expectation of the shifted Half-Normal distribution.

### G.1.3. NUMBER TOKEN LOSS (NTL)

Number Token Loss (NTL) (Zausinger et al., 2025) is an auxiliary regression objective designed to improve numerical predictability of autoregressive language models. Unlike standard CE, which treats numbers as independent nominal tokens, NTL penalizes the numerical distance between the predicted distribution and the ground truth of each numeric token. We implement the two primary variants proposed by the authors:

**NTL-MSE.** This variant treats the model's output as a continuous expectation. It minimizes the MSE between the numerical value of the ground truth token and the expected numerical value derived from the predicted probability distribution. Let $V(j)$ denote the numerical value of token $j$, $\omega_t$ be ground truth numeric token at step $t$, and $\mathcal{N}$ be the set of indices for number tokens:

$$\mathcal{L}_{\text{NTL}-\text{MSE}} = \frac{1}{K} \sum_{t=1}^{K} \left( V(\omega_t) - \sum_{j \in \mathcal{N}} p_t^j \cdot V(j) \right)^2,$$

where $p_t^j$ denotes the predicted probability assigned to token $j$ at step $t$, , and $K$ represents the total number of numeric tokens in the sequence.

**NTL-WAS.** To address potential optimization issues in MSE (e.g., non-unique minima), NTL-WAS minimizes the Wasserstein-1 distance. For a one-hot ground truth distribution, this simplifies to the expected absolute difference:

$$\mathcal{L}_{\text{NTL}-\text{WAS}} = \frac{1}{K} \sum_{t=1}^{K} \sum_{j \in \mathcal{N}} p_t^j \cdot |V(\omega_t) - V(j)|.$$

**Implementation Details.** The model is optimized using a joint objective: $\mathcal{L} = \mathcal{L}_{\text{CE}} + \lambda \cdot \mathcal{L}_{\text{NTL}}$, where we set the hyperparameter $\lambda$ to 0.3. The auxiliary loss is computed exclusively on numerical tokens, while non-numerical tokens are masked out.

### G.1.4. DIST$^2$ LOSS

DIST$^2$ Loss (Chung et al., 2026) introduces a distance-aware framework that integrates metric relationships directly into the target distribution of discrete autoregressive models. DIST$^2$ constructs a soft, categorical target distribution $p_d$ based on the inherent distance metric $d$ between the ground truth token $\omega$ and vocabulary tokens $j$.

The target distribution is modeled as a discretized exponential family distribution, where tokens closer to the ground truth in the metric space are assigned higher probabilities. This is controlled by a temperature parameter $T$:

$$p_d(j \mid \omega) = \frac{\exp(-d(j, \omega)/T)}{\sum_{j' \in \mathcal{N}} \exp\left(-d\left(j', \omega\right)/T\right)},$$

where we set $T = 1.0$ and the distance metric $d$ as Euclidean distance. The objective minimizes the KL divergence between this distance-aware target distribution and the model's predicted distribution $p_\theta$:

$$\mathcal{L}_{\text{DIST}^2} = \sum_{t=1}^{K} \sum_{j \in \mathcal{N}} p_d\left(j \mid \omega_t\right) \log \frac{p_d\left(j \mid \omega_t\right)}{p_\theta\left(j \mid \omega_{<t}\right)},$$

where $K$ denotes the sequence length, $\omega_t$ represents the ground truth token at step $t$, and $\mathcal{N}$ is the set of indices corresponding to number tokens in the vocabulary.

In this work, we adopt a joint training strategy that combines the standard CE loss ($\mathcal{L}_{\text{CE}}$) with the $\text{DIST}^2$. The final optimization objective is formulated as: $\mathcal{L} = \mathcal{L}_{\text{CE}} + \lambda \cdot \mathcal{L}_{\text{DIST}^2}$. Following the default configuration of the original paper, we set the weighting hyperparameter $\lambda$ to 0.1. The auxiliary loss is computed exclusively on numerical tokens, while non-numerical tokens are masked out.

## G.2. Model Architecture

Consistent with the decoding-based regression paradigm, all models employed in this work utilize an encoder-decoder framework. The specific architectural choices are tailored to the input modalities of the respective tasks.

### G.2.1. TABULAR REGRESSION

**Decoding-based Regressor Architecture.** Following (Song & Bahri, 2025), we utilize a hybrid architecture composed of an MLP encoder and a Transformer decoder to handle numerical feature inputs.

- **Encoder:** The encoder is implemented as a Multi-Layer Perceptron (MLP) to project continuous input features into the latent space. It consists of multiple hidden layers with Rectified Linear Unit (ReLU) activation functions. The input layer dynamically adjusts to the dimensionality of the feature vector $\mathbf{x}$, while the final linear layer projects the representation to the model dimension $d_{model}$ to align with the decoder.

- **Decoder:** The decoder follows a standard Transformer architecture (Vaswani et al., 2017) to autoregressively generate the target token sequence. It comprises a stack of decoder layers with multi-head attention mechanisms to balance computational efficiency with modeling capacity.

- **Tokenizer Configuration:** We employ the normalized tokenization strategy. The specific configurations, including the digit base $B$ and the number of digits $K$, are determined via hyperparameter search to adapt to different datasets.

**Baseline Regressor Architectures.**

- **Pointwise Head (MLP):** To ensure a fair comparison, we scale the Pointwise regression baseline to have a parameter count comparable to or larger than the decoding-based model. It is implemented as a deep MLP with ReLU activations.

- **Riemann Head:** For the Riemann head baseline, we discretize the target space into $K$ bins within the support range $[-3, 3]$ (applied to normalized targets). The backbone encoder follows the same MLP architecture as the Pointwise Head.

**Hyperparameter Search.** To eliminate manual bias and ensure fair comparison, we employ **Optuna** (Akiba et al., 2019) to automatically search for the optimal hyperparameters for both the decoding-based base model and the baseline regressors. For each method on each task, we run the search for 25 trials. Importantly, the hyperparameter search for the decoding framework is exclusively applied to the base model. All subsequent RL-based approaches (including GenRe$^2$) are directly initialized from this optimized base model and strictly inherit its hyperparameter configuration without further tuning.

The search space for the Decoding-based base model is detailed in Table 4. The search jointly optimizes the learning rate, tokenization settings (base and precision), and model architecture (dimensions and depth). Note that the number of hidden layers in the MLP encoder is coupled with the number of decoder layers to maintain architectural symmetry during the search.

The search space for the Baseline Models (Pointwise & Riemann MLP) is provided in Table 5.

### G.2.2. CODE METRIC REGRESSION

The pretrained model provided by (Akhauri et al., 2026) is an encoder-decoder model, where the encoder is a pretrained T5Gemma (Zhang et al., 2025a;b) encoder and the decoder is a standard Transformer decoder trained from scratch with the IEEE tokenizer (IEEE, 2019) with digit base $B = 10$, exponent length $E = 3$, and mantissa length $M = 5$. Since Akhauri et al. (2026) trained the model with the encoder frozen, we also freeze the encoder and finetune the decoder with respective strategies.

*Table 4.* Hyperparameter search space for the Decoding-based Base Model (Optuna, 25 trials).

| Hyperparameter | Search Range / Values |
|---|---|
| *Optimization* | |
| Learning Rate | Log-uniform $[1 \times 10^{-5}, 1 \times 10^{-4}]$ |
| *Tokenization* | |
| Digit Base ($B$) | $\{2, 4, 6, 8, 10\}$ |
| Digits Precision ($K$) | $\{4, 6, 8\}$ |
| *Model Architecture* | |
| Model Dimension ($d_{model}$) | $\{128, 256, 512\}$ |
| Attention Heads ($H$) | $\{4, 8\}$ |
| Feedforward Dimension ($d_{ff}$) | $\{512, 1024, 2048\}$ |
| Decoder Layers ($L_{dec}$) | Integer $[1, 5]$ |
| Encoder Hidden Dim | $\{128, 256, 512, 1024\}$ |
| Encoder Layers | Coupled with $L_{dec}$ |

*Table 5.* Hyperparameter search space for Baseline Models (Pointwise & Riemann).

| Hyperparameter | Search Range / Values |
|---|---|
| Learning Rate | Log-uniform $[1 \times 10^{-5}, 1 \times 10^{-2}]$ |
| Hidden Dimension | Integer $[1, 8]$ |
| Number of Layers | Integer $[64, 512]$ |
| Dropout | Float $[0, 0.5]$ |
| Weight Decay | Log-uniform $[1 \times 10^{-6}, 1 \times 10^{-3}]$ |
| *Riemann Only* | |
| Number of Bins | $\{16, 64, 256, 1024, 4096, 16384\}$ |

### G.2.3. GENERATIVE REWARD MODELS

`Mistral-7B-Instruct-v0.2` (Jiang et al., 2023) is an instruct fine-tuned 7-billion-parameter language model based on the transformer architecture. It inherits the structural foundations of its predecessor, featuring a model dimension of 4096, 32 layers, and a hidden dimension of 14336. The model retains the grouped-query attention mechanism with 32 query heads and 8 key-value heads. However, distinct from v0.1, this version removes the sliding window attention mechanism and significantly extends the context window from 8192 to 32768 tokens. To support this extended context, it utilizes a RoPE theta base of $1 \times 10^6$, while maintaining a vocabulary size of 32000.

### G.3. Data Processing

We apply specific data processing strategies according to the input modality and the regression head employed. All statistics used for normalization are computed exclusively from the training set to prevent data leakage.

**Input Processing.** The preprocessing of input features $\mathbf{x}$ depends on the task type:

- **Tabular Regression:** Since the inputs are numerical vectors, we apply standard z-score normalization to enhance numerical stability following the common protocol in tabular research (Liu et al., 2025a; Ye et al., 2024a):

$$\mathbf{x} \leftarrow \frac{\mathbf{x} - \boldsymbol{\mu}_x}{\boldsymbol{\sigma}_x},$$

where $\boldsymbol{\mu}_x$ and $\boldsymbol{\sigma}_x$ denote the coordinate-wise mean and standard deviation of the training inputs, respectively.

- **Code Metric Regression:** The inputs for these tasks are raw textual code. Consequently, we feed the text directly into the encoder without applying any additional numerical normalization.

**Target Processing.** The processing of target values $y$ is determined by the specific regression head and tokenization scheme:

- **Non-Decoder Heads (Pointwise & Riemann):** For these baselines, we standardize the targets using z-score normalization:

$$y \leftarrow \frac{y - \mu_y}{\sigma_y},$$

  where $\mu_y$ and $\sigma_y$ represent the mean and standard deviation of the training targets, respectively.

- **Decoder Heads:** For decoding-based regression, the strategy varies by tokenization scheme:

  - *P10 and IEEE Tokenization:* These schemes are designed to represent the raw numbers directly via scientific notation. Therefore, we do not apply any normalization and train the model to regress the raw target values.
  - *Normalized Tokenization:* This scheme requires targets to be bounded within a fixed interval. We adopt a two-stage scaling strategy: targets are first standardized via z-score, followed by Min-Max scaling. The transformation is defined as:

$$y' = \frac{y - \mu_y}{\sigma_y}, \quad y \leftarrow \frac{y' - \min(y')}{\max(y') - \min(y')}.$$

### G.4. Implementation Details

**Training Hyperparameters.** The token level term of GenRe[2] was set to different value for these tasks. For tabular regression tasks, we use 0.05, while changing it to 0.1 for code metric regression tasks and 0.001 for generative reward models. Optimization is performed using the AdamW optimizer. The specific configurations for each domain are as follows:

- **Tabular Regression:** For decoding-based methods, we use a batch size of 128 and an initial learning rate of $5 \times 10^{-5}$. The learning rate follows a cosine annealing schedule with 100 warmup steps and a minimum decay ratio of 0.1. The Base Model pretraining (CE) is conducted for 200 epochs, while the proposed Policy Gradient optimization runs for 100 epochs. For the baseline regression heads (MLP), we utilize the default training framework and hyperparameters provided by the TALENT benchmark.

- **Code Metric Regression:** We employ a batch size of 8 with a lower initial learning rate of $1 \times 10^{-6}$, and train for a total of 20 epochs, to preserve the pre-trained knowledge of the backbone during RL training. The model is fine-tuned for a total of 20 epochs before training when applying RL. For the token-level baselines(CE, NTL and DIST[2]), we use a batch size of 16 with the learning rate of $5 \times 10^{-6}$.

- **Generative Reward Models:** We use a batch size of 64 with the learning rate of $1 \times 10^{-5}$. The model is fine-tuned with LoRA, setting rank 8, alpha 16 as well as the maximum sequence length of 2048.

**Model Selection.** To ensure optimal performance and fair comparison, we employ different checkpoint selection strategies based on the training objective. For all standard regression models (including baselines and the pretrained base model), we select the checkpoint that achieves the lowest validation loss. Conversely, for GenRe[2], we select the checkpoint that yields the highest mean rewards on the validation set.

**Inference and Rollout Settings.** During the reinforcement learning phase (rollout), we employ stochastic sampling to encourage exploration.

- **Rollout:** We set the sampling temperature to 1.0. The number of samples generated per input is set to 16 for Tabular Regression , 4 for Code-to-Metric Regression and 4 for Generative Reward Models.

- **Evaluation:** For final inference on the test set, we maintain the temperature at 1.0 and aggregate predictions using the median of the generated candidates. The sampling budget is increased to 128 samples for Tabular Regression, 64 samples for Code-to-Metric Regression and 1 sample for Generative Reward Model to ensure robust estimation.

**RL Settings** We provide the specific hyperparameter settings and reward shaping techniques for the RL stage across different tasks below.

- **Tabular Regression:** For the tabular tasks, we employ both ReMax (Li et al., 2024) and GenRe$^2$ to finetune the base model. To ensure a fair comparison with the Pointwise and Riemann baselines (which operate on standardized targets), we define the mapping $\psi$ as a z-score standardization: $\psi(y) = \frac{y-\mu_y}{\sigma_y}$, where $\mu_y$ and $\sigma_y$ are the mean and standard deviation of the training set. The reward is calculated as the negative MSE on this standardized scale. We use the AdamW optimizer (Loshchilov & Hutter, 2019) for optimization with an initial learning rate of $1 \times 10^{-5}$.

- **Code Metric Regression:** Unlike tabular data, code metrics often exhibit heavy tails. As shown in Figure 8 in Appendix H.6, z-score distributions retain extreme values that can destabilize RL. Therefore, we use a **Quantile Transformation** to map targets to a standard Gaussian distribution. The number of quantiles is adaptively set to $\text{clip}(\lfloor N_{\text{train}}/30 \rfloor, 10, 1000)$, where $N_{\text{train}}$ is the training set size. To further prevent instability from extreme outliers during the early stages of RL, we clip the reward with a minimum threshold of $-50$:

$$R(\tau) = \max\left(-(\psi(\hat{y}) - \psi(y))^2, -50\right)$$

- **Generative Reward Models:** To ensure format and accuracy, we use a composite reward: $R(\tau) = 1.0 - (\hat{y} - y)^2$ for valid outputs, with a $-20.0$ penalty for regex failures. We apply a KL penalty ($\beta = 0.1$) against the teacher model to employ OPD. Training uses $G = 4$ samples with a greedy baseline and a learning rate of $1 \times 10^{-5}$.

### G.5. Evaluation.

Regarding tabular regression tasks, we compare all methods on a suite of regression metrics, including RMSE, R$^2$, and Spearman's Rank Correlation. For decoding-based methods, following (Akhauri et al., 2025), we directly sample from the model's output distribution with temperature 1.0 to generate $m = 128$ candidate solutions, and aggregate them via both mean and median. As for datasets in code metric regression and generative reward models, we use spearman's rank correlation.

## H. Additional Experiment

### H.1. Results for Code Metric Regression on High-level Source Code

*Table 6.* Results for code metric regression on CodeNet datasets comparing Spearman's Rank Correlation. We train models and report results as the average over 5 random seeds. The best and runner-up results are **bolded** and underlined, respectively. The row shaded in gray indicates our proposed method.

| Method | C | C# | C++ | Go | Haskell | Java | JavaScript | Kotlin | Python | Ruby | Rust |
|---|---|---|---|---|---|---|---|---|---|---|---|
| Base Model | **0.753**$_{\pm0.000}$ | 0.561$_{\pm0.000}$ | **0.761**$_{\pm0.000}$ | **0.680**$_{\pm0.000}$ | 0.605$_{\pm0.000}$ | **0.497**$_{\pm0.000}$ | 0.406$_{\pm0.000}$ | 0.615$_{\pm0.000}$ | **0.651**$_{\pm0.000}$ | **0.502**$_{\pm0.000}$ | 0.619$_{\pm0.000}$ |
| +CE | 0.682$_{\pm0.015}$ | 0.546$_{\pm0.009}$ | 0.685$_{\pm0.014}$ | 0.660$_{\pm0.002}$ | 0.615$_{\pm0.003}$ | 0.444$_{\pm0.011}$ | 0.453$_{\pm0.014}$ | 0.598$_{\pm0.015}$ | 0.530$_{\pm0.017}$ | 0.455$_{\pm0.018}$ | 0.585$_{\pm0.024}$ |
| +NTL-WAS | 0.683$_{\pm0.010}$ | 0.545$_{\pm0.007}$ | 0.697$_{\pm0.011}$ | 0.657$_{\pm0.002}$ | 0.607$_{\pm0.015}$ | 0.441$_{\pm0.004}$ | 0.448$_{\pm0.011}$ | 0.600$_{\pm0.019}$ | 0.528$_{\pm0.011}$ | 0.456$_{\pm0.018}$ | 0.603$_{\pm0.008}$ |
| +NTL-MSE | 0.677$_{\pm0.010}$ | 0.528$_{\pm0.008}$ | 0.705$_{\pm0.027}$ | 0.641$_{\pm0.013}$ | 0.605$_{\pm0.018}$ | 0.456$_{\pm0.007}$ | 0.459$_{\pm0.010}$ | 0.628$_{\pm0.005}$ | 0.557$_{\pm0.008}$ | 0.447$_{\pm0.003}$ | 0.593$_{\pm0.017}$ |
| +DIST$^2$ | 0.672$_{\pm0.019}$ | 0.552$_{\pm0.004}$ | 0.690$_{\pm0.003}$ | 0.664$_{\pm0.004}$ | **0.624**$_{\pm0.005}$ | 0.447$_{\pm0.010}$ | 0.464$_{\pm0.014}$ | 0.600$_{\pm0.005}$ | 0.509$_{\pm0.032}$ | 0.465$_{\pm0.010}$ | 0.599$_{\pm0.027}$ |
| **+ReMax** | 0.681$_{\pm0.025}$ | 0.559$_{\pm0.014}$ | 0.695$_{\pm0.006}$ | 0.596$_{\pm0.050}$ | 0.615$_{\pm0.002}$ | 0.475$_{\pm0.016}$ | 0.500$_{\pm0.016}$ | 0.661$_{\pm0.010}$ | 0.565$_{\pm0.016}$ | 0.388$_{\pm0.041}$ | 0.627$_{\pm0.015}$ |
| **+GenRe$^2$** | 0.689$_{\pm0.017}$ | **0.571**$_{\pm0.013}$ | 0.705$_{\pm0.003}$ | 0.663$_{\pm0.012}$ | 0.621$_{\pm0.003}$ | 0.479$_{\pm0.011}$ | **0.503**$_{\pm0.004}$ | **0.663**$_{\pm0.006}$ | 0.565$_{\pm0.025}$ | 0.431$_{\pm0.010}$ | **0.634**$_{\pm0.014}$ |

We extend our analysis to Spearman's rank correlation on CodeNet across multiple languages. Despite the strong performance of the base model, GenRe$^2$ achieves the best or runner-up results in most settings. Notably, ReMax also exhibits comparable stability when contrasted with token-level methods.

### H.2. Impact of Different Tokenization Schemes

We additionally evaluate the performance of GenRe$^2$ across different output tokenization schemes, i.e., P10 (Charton, 2022) and IEEE floating-point representations (IEEE, 2019). Given that P10 and IEEE tokenization could yield outlier predictions due to the model's hallucinations (Song et al., 2024; Song & Bahri, 2025), we also include an outlier filtering strategy for the generated candidates. From the R$^2$ and Spearman's rank correlation results in Table 7, we find that GenRe$^2$ consistently outperforms the base model under different output tokenizations, except for mean R$^2$ on P10, median and mean R$^2$ on IEEE, showing the robustness of GenRe$^2$. Besides, it is worth mentioned that tokenization with unlimited output range, e.g.,

*Table 7.* Ablation study on different output tokenization schemes comparing $R^2$ and Spearman's Rank Correlation. The results are reported as the average over 5 random seeds across 100 TALENT regression tasks. The best results are **bolded**. Rows shaded in gray highlight the ReMax results for direct comparison against CE.

| Tokenization | Metric | Method | Aggregation Strategy | | | |
| --- | --- | --- | --- | --- | --- | --- |
| | | | **Median** | **Median + Filter** | **Mean** | **Mean + Filter** |
| Norm. | $R^2$ | Base | 0.6215 | / | 0.6329 | / |
| | | GenRe$^2$ | **0.6588** | / | **0.6624** | / |
| | Rank Corr. | Base | 0.7668 | / | 0.7618 | / |
| | | GenRe$^2$ | **0.7898** | / | **0.7743** | / |
| P10 | $R^2$ | Base | $-2.46 \times 10^9$ | 0.5874 | $\mathbf{-6.75 \times 10^{24}}$ | 0.6102 |
| | | GenRe$^2$ | **0.6100** | **0.6173** | $-7.73 \times 10^{20}$ | **0.6328** |
| | Rank Corr. | Base | 0.7630 | 0.7630 | 0.7161 | 0.7605 |
| | | GenRe$^2$ | **0.7861** | **0.7861** | **0.7666** | **0.7741** |
| IEEE | $R^2$ | Base | $\mathbf{-2.95 \times 10^{17}}$ | 0.5947 | $\mathbf{-2.76 \times 10^{17}}$ | 0.6199 |
| | | GenRe$^2$ | $-4.43 \times 10^{17}$ | **0.6048** | $-4.44 \times 10^{17}$ | **0.6226** |
| | Rank Corr. | Base | 0.7652 | 0.7652 | 0.7389 | 0.7619 |
| | | GenRe$^2$ | **0.7836** | **0.7841** | **0.7727** | **0.7759** |

*Table 8.* RMSE, $R^2$ and Spearman's Rank Correlation results over a single random seed on 100 TALENT regression tasks. The best and runner-up results are **bolded** and underlined, respectively. Rows shaded in gray denote the proposed methods.

| Method | RMSE ↓ | $R^2$ ↑ | Rank Corr. ↑ |
| --- | --- | --- | --- |
| GRPO | 0.5165 | 0.6290 | 0.7711 |
| REINFORCE++ | 0.5032 | **0.6490** | 0.7733 |
| ReMax | **0.5028** | **0.6490** | **0.7736** |

P10 or IEEE, is easier to produce outlier values, resulting in poor $R^2$. However, such tokenization schemes remain higher Spearman's rank correlation, implicitly capturing the relationship between numbers. We also observe that GenRe$^2$ mitigates outliers in most cases, even obtaining positive median $R^2$ for P10, but it cannot eliminate the hallucinations. Reducing the hallucinations for unbounded tokenization is still a crucial future work for decoding-based regression (Song et al., 2024; Song & Bahri, 2025).

### H.3. Impact of Different RL Backbone

To investigate the impact of the optimization algorithm, we compare ReMax against two advantage normalization methods: standard GRPO, which employs local group-level normalization, and REINFORCE++, which utilizes global batch-level normalization (Hu et al., 2025).

The results in Table 8 show that REINFORCE++ performs comparably to ReMax, and both significantly outperform GRPO.

This comparison isolates the performance degradation to GRPO's local normalization. By forcing each group's advantage variance to 1, GRPO squashes catastrophic and minor errors into the same scale, destroying inter-group magnitude differences. In contrast, REINFORCE++ divides rewards by a global batch standard deviation. This shared scalar preserves relative sample weights, ensuring severe mistakes still yield larger gradient updates. This confirms that preserving the absolute error magnitude is essential for decoding-based regression.

### H.4. Analysis of Catastrophic Outliers

Unbounded tokenization (e.g., P10, IEEE) is highly sensitive to high-order digit errors, which often lead to catastrophic outliers (Golkar et al., 2023). As shown in Table 9, GenRe$^2$ achieves the lowest outlier ratio, following the trend:

*Table 9.* Proportion of catastrophic outliers during rollouts. Unlike the CE baseline, sequence-level RL (ReMax and GenRe$^2$) explicitly penalizes massive numerical errors, significantly mitigating highly erroneous predictions. GenRe$^2$ achieves the most substantial stability improvement, reducing the outlier ratio by 64.28% compared to the CE baseline. This demonstrates the framework's effectiveness in prioritizing absolute error reduction during the optimization process.

| metrics | CE | ReMax | GenRe$^2$ |
|---|---|---|---|
| Outlier Ratio of Rollouts | 4.461% | 2.443% | **1.594%** |
| Relative Reduction (w.r.t. CE) | - | 45.24% | **64.28%** |

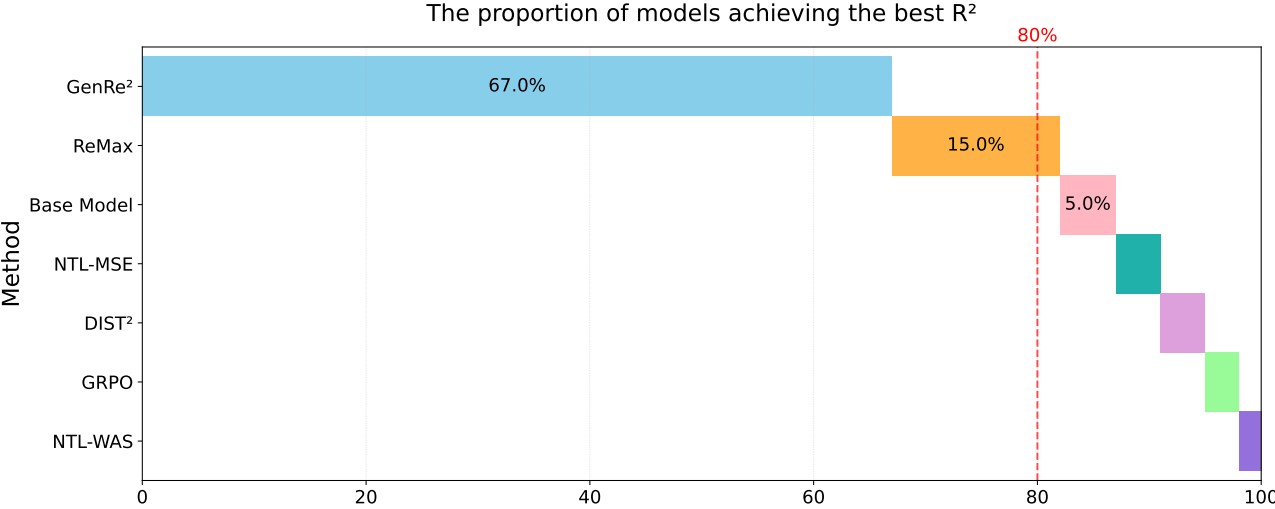

*Figure 7.* The proportion of models achieving the best R$^2$. The length of each bar represents the proportion of the 100 datasets (in which a given method achieved the highest R$^2$) on the TALENT benchmark. Note that all models utilized a normalized tokenizer with searched parameters

CE $>$ ReMax $>$GenRe$^2$. This confirms that directly optimizing continuous numerical error via RL penalizes large deviations more effectively than token-level CE, with dense expert guidance further enhancing the model's global magnitude awareness.

### H.5. Robustness of GenRe$^2$ Across Different Tokenizer Bases

We further investigate the impact of the tokenizer's base parameter on model ranking within the TALENT benchmark. As shown in Figures 7, GenRe$^2$ consistently achieves the highest R$^2$ score on the majority of the 100 datasets, regardless of the base selected. Specifically, GenRe$^2$ maintains a dominant position, securing the best performance across all configurations. This analysis confirms that GenRe$^2$ performs consistently well across different tokenizer settings, showing that it does not require specific tuning of the base parameter to achieve best results.

### H.6. Visualization of Target Normalization for Code Metric Regression

In this subsection, we visualize the target distribution under different normalization strategies mentioned in Section 4.2.1. As shown in Figures 8 and 9, we can observe that both on the APPS Leetcode and the Triton Kernel Latency dataset, the z-score standardization exhibits sharp distribution and is prone to outliers, while the quantile normalization based on Gaussian delivers a smooth one.

### H.7. Impact of GenRe$^2$ Finetuning

Figure 10 evaluates the distributional impact of GenRe$^2$ finetuning. Regarding entropy dynamics (Figure 10a), vanilla ReMax suffers a 67.4% entropy drop, indicating severe mode collapse. In contrast, GenRe$^2$ maintains significantly higher entropy (29.5% drop), demonstrating that token-level expert supervision preserves generative diversity and prevents RL over-optimization. Additionally, across 100 TALENT datasets (Figure 10b), GenRe$^2$ consistently achieves lower Wasserstein-1

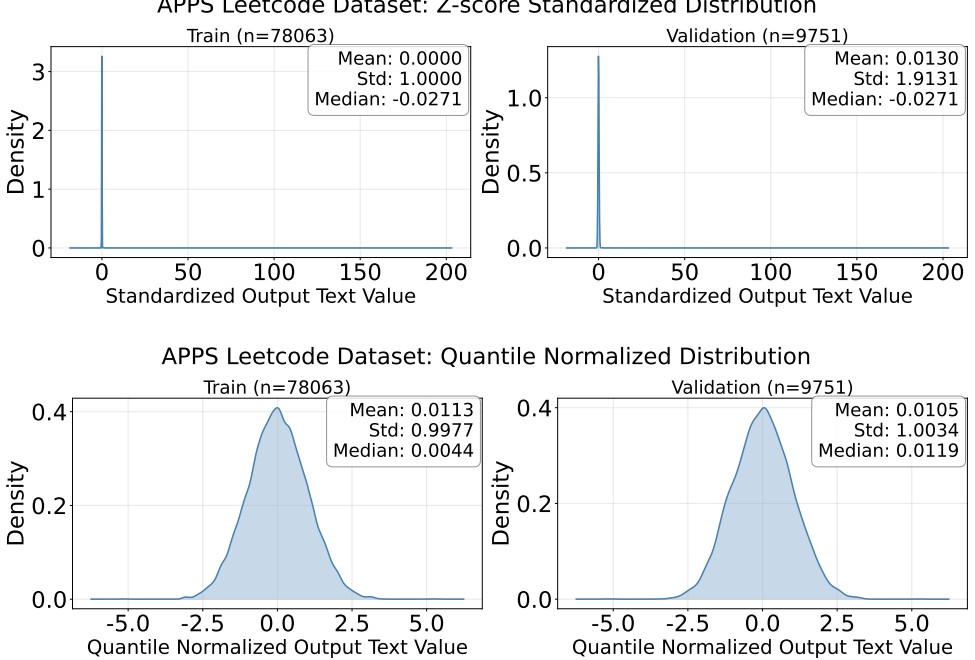

*Figure 8.* Comparison of target value distributions on the APPS Leetcode Dataset across training and validation sets. **Top row**: Z-score standardization results in distributions with heavy tails and extreme outliers in both the training (left) and validation (right) splits. **Bottom row**: In contrast, quantile normalization effectively transforms the target values into a well-formed standard normal distribution consistently across both subsets.

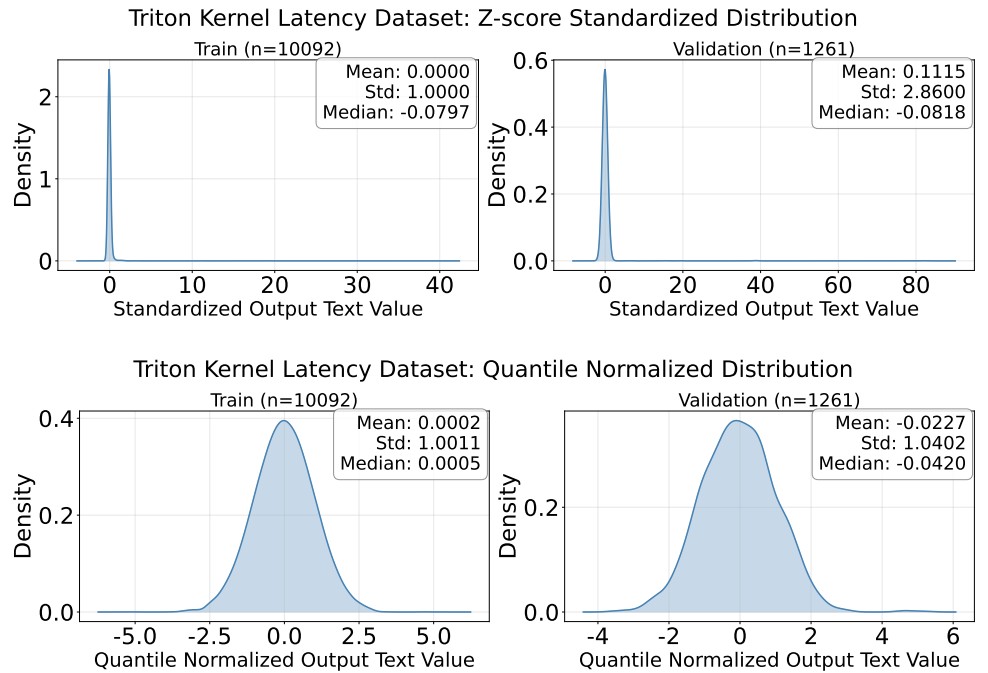

*Figure 9.* Comparison of target value distributions on the Triton Kernel Latency Dataset across training and validation sets. **Top row**: Z-score standardization results in distributions with heavy tails and extreme outliers in both the training (left) and validation (right) splits. **Bottom row**: In contrast, quantile normalization effectively transforms the target values into a well-formed standard normal distribution consistently across both subsets.

distances than the base model, confirming its superior approximation of the target distribution.

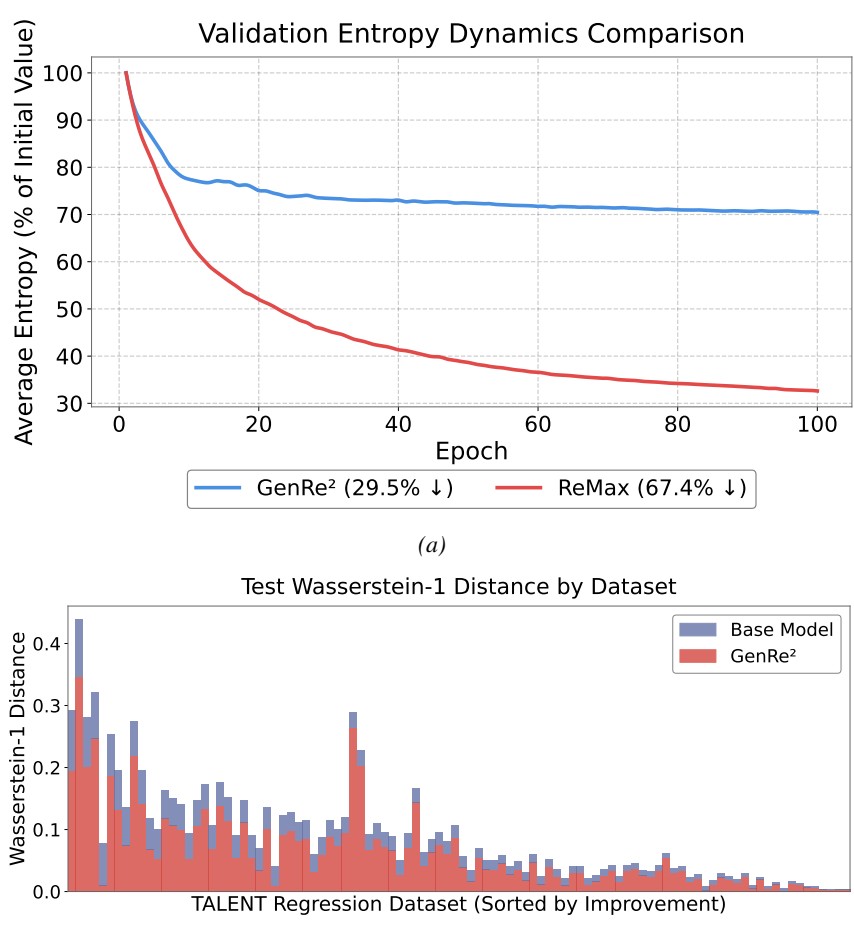

*Figure 10.* Impact of GenRe$^2$ finetuning on output distribution. **(a)** Comparison of entropy evolution during training. Vanilla ReMax (red) suffers from a drastic drop in entropy, indicating severe mode collapse. In contrast, GenRe$^2$ (blue) maintains a consistently higher entropy, demonstrating that token-level expert supervision preserves generative diversity by preventing the over-exploitation of pure RL. **(b)** Visualization of the test Wasserstein-1 distance (lower is better) across 100 regression datasets of TALENT benchmark, where GenRe$^2$ (red) consistently achieves lower distances compared to the base model (blue), demonstrating better approximation towards the ground truth target.

### H.8. Ablation of Non-uniform Token Weighting

To investigate whether the performance gap between token-level methods and GenRe$^2$ can be closed by simply prioritizing significant digits, we conduct ablation experiments using a non-uniform token weighting scheme. Specifically, we implement a smoothed exponential weighting function $w_k = \log(\text{base}^k + 1)$, where $k$ denotes the positional significance of the digit (higher $k$ for higher-order bits). To ensure a fair comparison, these weights are normalized by the sum of weights and then multiplied by the total sequence length $K$. This procedure ensures that the overall gradient update magnitude remains identical to that of the standard CE loss.

As shown in Table 10 and 11, while applying decreasing weights to CE or token-level guidance brings marginal improvements, the gains are significantly smaller than those achieved by sequence-level RL optimization in GenRe$^2$. This result reinforces our core premise that the primary bottleneck in decoding-based regression is the lack of global, sequence-level awareness, which cannot be fully resolved by local token reweighting.

### H.9. Extended Comparison with NTL on Original Task Settings

To ensure a rigorous comparison with the Number Token Loss (NTL) baseline, we conduct an additional experiment strictly following the task configuration and the T5-small architecture as used in Zausinger et al. (2025). Table 12 summarizes the results on this representative task. While NTL provides substantial improvements over the standard CE baseline, GenRe$^2$ achieves further performance gains.

*Table 10.* RMSE, $R^2$ and Spearman's Rank Correlation results over 1 random seed on 100 TALENT regression tasks. The best and runner-up results are **bolded** and underlined, respectively. The rows shaded in gray indicate our proposed methods. The · Reweight variants employ a smoothed exponential weighting scheme to prioritize earlier tokens. While positional reweighting yields marginal improvements, the substantial performance leap in GenRe$^2$ is primarily driven by sequence-level RL optimization.

| Method | RMSE ↓ | $R^2$ ↑ | Rank Corr. ↑ |
|---|---|---|---|
| CE | 0.5223 | 0.6271 | 0.7670 |
| CE · Reweight | 0.5199 | 0.6297 | 0.7679 |
| CE · Reweight+NTL | 0.5245 | 0.6140 | 0.7668 |
| (CE+NTL) · Reweight | 0.5188 | 0.6302 | 0.7674 |
| GenRe$^2$ | 0.4928 | 0.6593 | **0.7798** |
| GenRe$^2$ · Reweight | **0.4922** | **0.6597** | 0.7783 |

*Table 11.* Results (i.e., Spearman's Rank Correlation) for code metric regression on Triton Kernel Latency. We train models and report results based on a single random seed due to limited training time. The best and runner-up results are **bolded** and underlined, respectively. The rows shaded in gray indicate our proposed methods. The · Reweight variants employ a smoothed exponential weighting scheme to prioritize earlier tokens. Token-level approaches remain substantially inferior to sequence-level RL methods, demonstrating the irreplaceable advantage of global sequence optimization.

| Method | Rank Corr. ↑ |
|---|---|
| CE | 0.5553 |
| CE · Reweight | 0.5687 |
| CE · Reweight+NTL | 0.5380 |
| (CE+NTL) · Reweight | 0.5382 |
| ReMax | 0.6276 |
| GenRe$^2$ | **0.6437** |
| GenRe$^2$ · Reweight | 0.6301 |

*Table 12.* Empirical comparison on the *r*Jokes dataset. (Evaluated using the T5 architecture and task settings from the original NTL study). Best and runner-up results are **bolded** and underlined, respectively. Shaded rows denote the proposed sequence-level RL approaches. While NTL provides substantial improvements over the static CE baseline, GenRe$^2$ achieves further consistent performance gains across all metrics.

| Method | RMSE ↓ | Spearman ↑ |
|---|---|---|
| CE | 3.715 | 0.35 |
| NTL | 3.24 | **0.4** |
| GenRe$^2$ | **2.96** | **0.4** |

We hypothesize this gap arises from the transition to dynamic supervision. Unlike NTL's reliance on static targets, GenRe$^2$ leverages on-policy rollouts, which potentially helps the model better self-correct numerical deviations encountered during inference.

### H.10. Analysis of Boundary Token Predictions

We investigate whether GenRe$^2$ introduces a general bias toward boundary tokens ('0' or '9'). As shown in Figure 11, while the frequency of boundary tokens increases for both methods, GenRe$^2$ exhibits a significantly larger increase following a prefix mismatch compared to a match. This confirms that boundary tokens are primarily utilized as a targeted corrective mechanism for numerical recovery rather than representing a static generative bias.

### H.11. Comparison with Standard RL Regularizers

We evaluate GenRe$^2$ against standard RL regularization techniques, specifically an explicit entropy bonus (ReMax-Entropy) and a Kullback-Leibler penalty against a reference model (ReMax-KL). As shown in Table 13, we measure Spearman's Rank Correlation on the APPS Leetcode and Triton Kernel Latency code metric regression tasks.

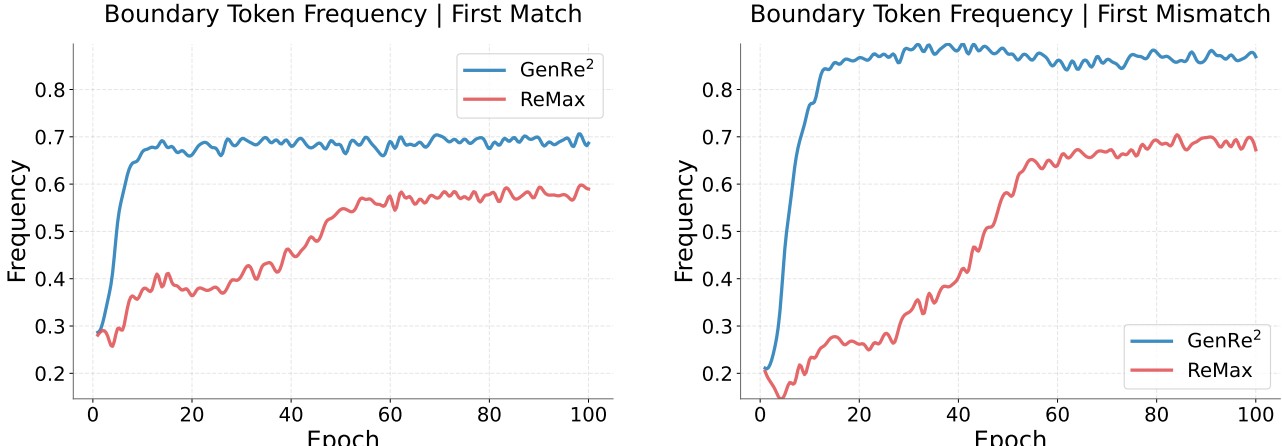

*Figure 11.* Boundary token frequency dynamics. Evaluated on the test set of a representative TALENT task (base-10, 3-digit configuration). (**Left**) First decoded digit matches the ground truth. (**Right**) First digit mismatch. GenRe$^2$ consistently generates more boundary tokens than the ReMax baseline, with a significantly larger increase under the mismatch condition. This confirms the active use of boundary tokens as a corrective mechanism to minimize absolute distance upon prefix deviation.

*Table 13.* Results (i.e., Spearman's Rank Correlation) for code metric regression on APPS Leetcode and Triton Kernel Latency. The best and runner-up results are **bolded** and underlined, respectively. The rows shaded in gray indicate our proposed methods.

| Method | APPS Leetcode | Triton Kernel Latency |
|---|---|---|
| ReMax-Entropy | 0.9740 | 0.6378 |
| ReMax-KL | 0.9726 | 0.6271 |
| ReMax | 0.9702 | 0.6276 |
| GenRe$^2$ | **0.9775** | **0.6437** |

While the addition of standard stabilizers (such as the entropy bonus) brings marginal improvements over vanilla ReMax, GenRe$^2$ consistently achieves the highest performance across both datasets. This demonstrates that the token-level expert guidance in GenRe$^2$ provides a more effective and targeted correction signal for numerical regression than generic RL regularizers.

### H.12. Computational Cost and Training Efficiency

We provide a detailed breakdown of the computational overhead associated with GenRe$^2$. Tables 14 compares the memory usage and wall-clock time per training step. While RL post-training is more resource-intensive than supervised fine-tuning due to autoregressive exploration (e.g., 16 samples per rollout), our expert oracle scoring adds negligible overhead.

To verify that the performance gains are not simply a result of increased compute, we conduct a wall-clock-aligned experiment. We extend the training of supervised baselines (NTL and $DIST^2$) to match the total computational budget of GenRe$^2$. As shown in Table 15, even with equivalent training time, the baselines' performance plateaus below that of GenRe$^2$. This confirms that the effectiveness of GenRe$^2$ originates from the RL-based sequence-level optimization itself.

#### H.12.1. Oracle Generation under Scientific-Notation Tokenizers.

We now describe the construction of the token-level expert oracle used by GenRe$^2$ for unnormalized scientific-notation tokenizers, including P10 and IEEE tokenization. The oracle is queried on-policy: given a ground-truth target $y$ and a model-generated prefix $\hat{\tau}_{<t}$, it returns an optimal next token $a_t^\star$ conditioned on the current prefix, rather than simply copying the next token in the canonical tokenization of $y$. This distinction is crucial because the sampled prefix may already contain errors; in such cases, the oracle should provide the best reachable correction under the prefix constraint.

For a tokenizer $\mathcal{T}$, each partial prefix $\hat{\tau}_{<t}$ induces a set of valid completions and therefore a reachable numerical set

$$\mathcal{S}_{\mathcal{T}}(\hat{\tau}_{<t}) = \{\text{Detok}_{\mathcal{T}}(\tau) : \tau \in \mathcal{C}_{\mathcal{T}}(\hat{\tau}_{<t})\},$$

where $\mathcal{C}_{\mathcal{T}}(\hat{\tau}_{<t})$ denotes all tokenizer-valid complete sequences that share the prefix $\hat{\tau}_{<t}$. Depending on the tokenizer and

the current position, this set may be a single interval, a finite union of intervals, or a discrete set of representable values. For example, in P10 tokenization, if the mantissa prefix has been partially fixed but the exponent remains unspecified, the reachable values can form multiple scale-separated intervals corresponding to different exponent choices.

The expert oracle selects the next token by reachable-set refinement:

$$a_t^\star = \arg \min_{a \in \mathcal{V}_t(\hat{\tau}_{<t})} D(\psi(y), \psi(\mathcal{S}_\mathcal{T}(\hat{\tau}_{<t}, a))),$$

where $\mathcal{V}_t(\hat{\tau}_{<t})$ is the set of syntactically valid next tokens, and $D(z, \mathcal{S}) = \inf_{v \in \mathcal{S}} |z - v|$ measures the distance from a scalar $z$ to a reachable set $\mathcal{S}$. In practice, this does not require enumerating all possible suffixes. The reachable set induced by each candidate token has a simple structured form: it is determined by the remaining free digits and exponent choices. Therefore, the oracle can be implemented by checking whether the target lies in the candidate reachable set and, if not, comparing only the closest boundary values of that set. Intuitively, if the target is still reachable under some candidate token, the oracle chooses a token that keeps the target reachable and further refines the representation. If the target is no longer reachable due to previous sampling errors, the oracle selects the candidate whose nearest reachable boundary is closest to the target.

For P10 tokenization, a scalar is represented by a sign token, $M$ mantissa digit tokens, and an exponent token:

$$\tau_{\text{P10}} = \langle s \rangle \langle d_1 \rangle \cdots \langle d_M \rangle \langle E_e \rangle.$$

The corresponding value is

$$\text{Detok}_{\text{P10}}(\tau) = s \cdot \left( \sum_{j=1}^M d_j B^{M-j} \right) \cdot B^e.$$

Before the exponent is fixed, a mantissa prefix does not correspond to a single continuous interval. Instead, it induces a union of intervals, one for each admissible exponent. Choosing the next mantissa digit refines each of these exponent-conditioned intervals simultaneously. The oracle therefore selects the digit whose induced union of reachable intervals contains the target if possible; otherwise, it selects the digit whose closest reachable boundary is nearest to the target. When the exponent token is generated, the mantissa has already been fixed by the sampled prefix, and the reachable set reduces to a finite collection of values indexed by the exponent vocabulary. The oracle then chooses the exponent whose detokenized value is closest to the target. For example, if the target is $52$ but the sampled prefix has already fixed the leading mantissa digit as $6$, the oracle does not force the canonical next digit of $52$. Instead, it chooses the continuation whose reachable set under the prefix beginning with $6$ is closest to $52$, e.g., preferring a continuation corresponding to $60$ over $62$ when these are the relevant reachable candidates.

For IEEE tokenization, we use a generalized floating-point format consisting of a value sign, an exponent sign, $E$ exponent digits, and $M$ mantissa digits:

$$\tau_{\text{IEEE}} = \langle s \rangle \langle s_e \rangle \langle e_1 \rangle \cdots \langle e_E \rangle \langle d_1 \rangle \cdots \langle d_M \rangle.$$

The detokenized value is

$$\text{Detok}_{\text{IEEE}}(\tau) = s \cdot \left( \sum_{j=1}^M d_j B^{1-j} \right) \cdot B^{s_e \cdot \sum_{k=1}^E e_k B^{E-k}}.$$

The oracle follows the same reachable-set refinement principle. The sign and exponent tokens determine the admissible order-of-magnitude regions, while the mantissa tokens refine the value within the selected scale. When the exponent prefix is still incomplete, each candidate exponent token induces a set of reachable scales and hence a union of mantissa-dependent intervals. The oracle chooses the token whose reachable set contains the target if possible, or otherwise whose closest boundary is nearest to the target. Once the exponent is fixed, the remaining mantissa prefix induces a single scale-specific interval, and each mantissa digit further subdivides this interval. This is particularly important for on-policy correction: if the model has already generated an incorrect exponent prefix, the oracle does not copy the canonical IEEE encoding of $y$, but instead provides the best mantissa continuation reachable under the erroneous magnitude.

This oracle construction is used only for the dense guidance term and does not change the sequence-level reward. The reward is still computed after the complete sampled sequence is detokenized. Therefore, GenRe$^2$ combines a global regression-aware reward with a tokenizer-specific local correction signal. Invalid next tokens are masked by $\mathcal{V}_t(\hat{\tau}_{<t})$, and invalid complete outputs, if any remain after detokenization, are assigned the same failure penalty as in the reward computation.

*Table 14.* Comparison of training costs across three different benchmarks. Across all tasks, RL-based methods (ReMax, GenRe$^2$) naturally incur higher resource consumption than the supervised baseline (CE) due to autoregressive exploration (parallel rollouts). Notably, GenRe$^2$ introduces negligible computational and memory overhead compared to the vanilla ReMax baseline.

| Method | Tabular Regression | | | Code Metric Regression | | | Generative Reward Models | | |
|---|---|---|---|---|---|---|---|---|---|
| | Peak Mem (GB) | Wall Time (s/epoch) | GPU Time (h/epoch) | Peak Mem (GB) | Wall Time (s/epoch) | GPU Time (h/epoch) | Peak Mem (GB) | Wall Time (s) | GPU Time (h) |
| CE | 0.15 | 0.075 | 0.000210 | 13.88 | 414.390 | 0.115108 | 11.72 | 205.594 | 0.057109 |
| ReMax | 2.95 | 0.962 | 0.000267 | 42.51 | 5299.368 | 1.472047 | 42.23 | 954.941 | 0.265261 |
| GenRe$^2$ | 2.96 | 0.993 | 0.000276 | 43.23 | 5571.267 | 1.547574 | 46.01 | 1015.711 | 0.282142 |

*Table 15.* Performance comparison under a matched wall-clock time budget. The training wall-clock time of the supervised baselines is extended to align with the time budget of the RL frameworks. Results (RMSE, R$^2$, and Spearman's Rank Correlation) are evaluated over a single random seed on 100 TALENT regression tasks. The best and runner-up results are **bolded** and underlined, respectively. Methods shaded in gray indicate our proposed approaches. Despite additional training time for baselines, GenRe$^2$ consistently maintains superior performance across all metrics, demonstrating that its advantages stem from effective optimization rather than increased training duration.

| Method | RMSE ↓ | R$^2$ ↑ | Rank Corr. ↑ |
|---|---|---|---|
| DIST$^2$ | 0.5300 | 0.6284 | 0.7745 |
| NTL-MSE | 0.5301 | 0.6283 | 0.7692 |
| NTL-WAS | 0.5337 | 0.6235 | 0.7737 |
| ReMax | 0.5154 | 0.6512 | 0.7720 |
| GenRe$^2$ | **0.4928** | **0.6593** | **0.7798** |

Consequently, this construction requires only basic arithmetic comparisons (i.e., simple `if-else` logic). The time complexity for determining the expert token at any decoding step is strictly $\mathcal{O}(1)$, introducing negligible computational overhead to the training process.

# I. Dataset Settings

- **APPS-LeetCode** (Hendrycks et al., 2021) is derived from the APPS benchmark, which contains 10K Python programming problems, 232.4K ground-truth solutions, and 131.7K test cases. We use this dataset to evaluate performance prediction for high-level Python programs, where the targets include execution latency and peak memory usage.

- **CodeNet** (Puri et al., 2021) is a large-scale multilingual code dataset. We use CodeNet-derived programs to evaluate memory prediction across diverse programming problems and languages.

- **Triton Kernel Latency** (Paliskara & Saroufim, 2025) focuses on estimating the execution latency of low-level Triton GPU kernels generated from PyTorch programs.

- **NAS-Bench-101** (Ying et al., 2019) records key performance metrics of neural architectures, including training, validation, and test accuracy, as well as training time and the number of trainable parameters.

- **NAS-Bench-201** (Dong & Yang, 2020) evaluates neural architectures using training, validation, and test accuracy and loss. It also provides computational metrics for each architecture, facilitating hardware-aware neural architecture search.

- **Amoeba** (Real et al., 2019) is a neural architecture search space defined by 8 operations and 5 nodes under a standardized macro-architecture. Its performance is typically characterized by the empirical distribution of top-1 test errors on CIFAR-10.

- **DARTS** (Liu et al., 2018b) defines a cell-based search space with 8 candidate operations and 4 intermediate nodes. The search space is evaluated by analyzing the distribution of error rates of sampled architectures.

- **ENAS** (Pham et al., 2018) defines a constrained cell-based search space with 5 candidate operations and 5 nodes. Its performance is evaluated through the distribution of CIFAR-10 accuracies of sampled architectures.

- **NASNet** (Zoph et al., 2018) constructs architectures from a search space with 13 candidate operations and 5 nodes, with varying depths and widths. The design space is evaluated by analyzing the empirical distribution of CIFAR-10 classification errors.

- **PNAS** (Liu et al., 2018a) builds neural architectures by progressively sampling and expanding cell structures with 8 candidate operations and 5 nodes. The design space is evaluated by comparing the cumulative distribution of model errors, often conditioned on parameter count and FLOPs.

## J. Settings of Training and Test Datasets

### J.1. Training Dataset

**Feedback Collection** (Kim et al., 2024a) is generated entirely by GPT-4. It serves as the training source and contains approximately 100K samples, including inputs, evaluations, and scores. This dataset provides a large-scale synthetic foundation for fine-tuning the model.

### J.2. Test Datasets

- **Feedback Bench** (Kim et al., 2024a) contains 1K responses with unique instructions and scoring rubrics that are not seen during training, enabling rigorous evaluation on unseen data.

- **FLASK** (Ye et al., 2024b) evaluates 2K responses generated by four models across 200 distinct prompts.

- **Vicuna Bench** (Chiang et al., 2023) is a single-turn dialogue benchmark. It is adapted for judge models by adding specific scoring rubrics to its 80 user instructions and evaluates 320 responses generated by multiple models.

- **MT-Bench** (Zheng et al., 2023) focuses on multi-turn dialogues across diverse tasks. It employs human-written rubrics for direct assessment and evaluates 320 responses from multiple models in complex conversational scenarios.

## K. Future Works

Based on our experimental results and analyses, there are many worthwhile directions for future exploration. Here we highlight some promising avenues for future research:

1. **Robust uncertainty calibration.** Our experiments in Section 4.4, together with prior works in RLVR (Zhu et al., 2025b; Yue et al., 2025; Matsutani et al., 2026) indicate that while RL is effective, it tends to over-sharpen the output distribution, which leads to uncalibrated prediction. However, this harms the uncertainty estimation capability delivered from pretraining (Song & Bahri, 2025; Bereket & Leskovec, 2025), which is important for response verification (Chen et al., 2025b) and real-world usage (Nguyen & Grover, 2022; Damani et al., 2026; Nguyen et al., 2024). Thus, an urgent need still exists for developing more robust and calibrated generative decoding-based regressors under the dynamics of RL-based post-training.

2. **Understanding the mechanism of RL update.** Recent works have identified the sparse weight update dynamics of RLVR (Shao & Wu, 2025; Mukherjee et al., 2025; Zhu et al., 2025a), which motivates for geometry-aware, parameter-efficient RL algorithm design. Although RL shows consistent improvements in decoding-based regression, the underlying mechanisms remain underexplored. It is worth studying further how RL changes the parameter and its association with regression metrics.

3. **Better RL algorithms.** Although RLVR algorithms (i.e., ReMax and GRPO used in this paper) show good capabilities, our analysis of the best@$k$ metric in Section 4.4 suggests that current algorithms have not fully explored the search space for decoding-based regression. Thus, techniques like entropy regularization (Cui et al., 2025), improving sampling efficiency (Karan & Du, 2026), and negative samples reinforcement (Zhu et al., 2025b) can be further explored.

4. **Combination with modern tabular regression structures.** In this paper, we mainly use MLP as the encoder for tabular regression. The generalization of decoding-based regression upon other prevalent tabular model structures (Yan et al., 2023; Chen et al., 2023; Gorishniy et al., 2021; Ye et al., 2025) and tabular foundation models (Zhang et al., 2025f; Hollmann et al., 2025; Grinsztajn et al., 2025; Zhang et al., 2025e) is worth further studied.

