# OpenReview forum: "Beyond Token-level Supervision: Unlocking the Potential of Decoding-based Regression via Reinforcement Learning"
_ICML.cc/2026/Conference — ICML 2026 regular_

### Official Review · Reviewer_MNE6 · 2026-03-11

**Soundness:** 3
**Presentation:** 3
**Significance:** 3
**Originality:** 3
**Overall Recommendation:** 4
**Confidence:** 3

**Summary:**

This paper addresses a fundamental limitation in decoding-based regression, where large language models predict continuous numerical values by generating discrete token sequences. The authors identify that standard token-level training (e.g., Cross-Entropy loss) fails to capture the global mathematical magnitude of the predicted numbers. To solve this, the paper proposes GenRe², a novel framework that reformulates the task as a Markov Decision Process. GenRe² utilizes a sequence-level reinforcement learning objective (specifically ReMax) to preserve absolute error magnitudes, while simultaneously injecting an on-policy, token-level "expert oracle" to provide dense guidance. This dual approach aims to align the model with true regression metrics and solve the temporal credit assignment problem. The method is evaluated across three diverse domains: tabular regression, code metric prediction, and generative reward modeling.

**Compliance With Llm Reviewing Policy:**

Affirmed.

**Final Justification:**

Most of my concerns have been addressed. I maintain my positive score.

**Key Questions For Authors:**

1. Regarding the critique of GRPO: Can you evaluate REINFORCE++ (which utilizes global advantage normalization) on your benchmarks? If REINFORCE++ performs well, does this alter your core argument that reward normalization inherently induces detrimental scale invariance, or does it isolate the issue strictly to GRPO's local normalization?
2. Regarding RL stability: Did you experiment with standard RL stabilizers such as a KL divergence penalty against a reference model, or an explicit entropy bonus during the ReMax update? How does GenRe² perform with these standard stabilizers compared to relying purely on dense expert guidance to prevent mode collapse?
3. Regarding computational overhead: Please provide a transparent breakdown of the computational costs. Specifically, what are the GPU-hours and wall-clock times for training? Most importantly, what is the exact training overhead (in time and memory) of GenRe² compared to the standard token-level CE baseline?

**Limitations:**

yes

**Strengths And Weaknesses:**

### Strengths:
1. Originality: The combination of sequence-level reinforcement learning with token-level expert guidance for numerical regression is a creative and highly practical solution to a well-documented misalignment in generative regression. The theoretical analysis demonstrating how ReMax preserves error magnitude—whereas standard GRPO eliminates it—is insightful.
2. Significance: The work tackles an increasingly important problem as LLMs are deployed for quantitative tasks. By proving efficacy across three distinct domains (tabular, code metrics, and reward modeling), the authors demonstrate that GenRe² is a versatile and general-purpose paradigm.
3. Presentation: The motivation is extremely clear. The paper effectively uses intuitive examples (e.g., the catastrophic error of predicting the most significant digit incorrectly) to justify the need for sequence-level rewards.

### Weaknesses:
1. Narrow Critique of Normalization: The paper dedicates significant effort to mathematically proving that GRPO’s reward normalization destroys error magnitude, concluding that ReMax is superior. While this critique of local batch normalization is mathematically sound and compelling, the scope is too narrow. The authors completely ignore global normalization alternatives. The industry has already recognized GRPO's local batch issues and proposed REINFORCE++ (which uses global advantage normalization). By omitting a comparison with REINFORCE++, the paper's broad claim that "normalization harms regression" remains unproven; it may be that only local normalization is harmful.
2. Unverified Mechanism for Preventing Mode Collapse: The paper claims dense expert guidance prevents mode collapse and preserves entropy. However, it lacks a baseline comparison against pure RL with standard stabilizers (e.g., a KL penalty or explicit entropy bonus) in the tabular and code metric tasks.
3. Opaque Computational Costs: Reinforcement learning significantly increases both code complexity and training overhead. However, the experimental details regarding computational costs are highly ambiguous. The paper fails to report critical training metrics such as GPU-hours, wall-clock time, or a direct comparison of the training overhead of GenRe² relative to the standard supervised Cross-Entropy (CE) baseline.

---

> ### Author Rebuttal · Authors · 2026-03-31
>
> Thanks for your valuable comments. The additional results can be found at https://anonymous.4open.science/api/repo/GenRe-re-3810/file/MNE6.pdf. Below please find our response.
>
> ### Q1 Evaluation of REINFORCE++ and reward normalization
>
> Thanks for bringing REINFORCE++ to our vision. We evaluated REINFORCE++ in Table R1, where it indeed performs comparably to ReMax. This isolates the undesirable scale invariance to GRPO's local normalization.
>
> The failure of GRPO stems from its group-level normalization, which mathematically forces the advantage variance of each group to be exactly 1. This squashes catastrophic errors and minor errors into the same advantage scale, destroying the inter-group magnitude differences.
>
> In contrast, REINFORCE++ utilizes batch-level normalization. Across a large batch (128 in our experiment), dividing all rewards by a global standard deviation merely applies a constant scalar.
> Crucially, this preserves the relative weights of each sample's contribution to the overall batch gradient (e.g., a -100 penalty remains 10x larger than -10).
> This is consistent with our core analysis between ReMax and GRPO: the key is to preserve cross-sample error magnitude. Since batch-level normalization applies a shared rescaling factor, it does not erase the relative difference between large-error and small-error samples, and thus still allows severe mistakes to produce larger gradient updates.
>
> We will add REINFORCE++ as an RL backbone and include this discussion in the revision.
>
> ### Q2 Comparison with standard RL stabilizers
>
> We thank the reviewer for this insightful question. We have compared GenRe² standard RL stabilizers, including both a KL penalty against a reference model and an explicit entropy bonus during the ReMax update.
>
> As presented in Table R2-R3, we compared GenRe² against baselines equipped with these standard stabilizers. The results clearly demonstrate that while KL and entropy penalties offer marginal stability improvements, GenRe², relying on dense expert guidance, consistently achieves the better performance.
>
> Here we theoretically explain why standard stabilizers behave differently from GenRe² in continuous numerical regression.
> Let $z_k$ denote the pre-softmax logit for a candidate token $k$, and $\pi_k = \frac{\exp(z_k)}{\sum_j \exp(z_j)}$ be its corresponding policy probability. The system entropy is defined as $H = -\sum_j \pi_j \log \pi_j$, and $\pi_{\text{ref},k}$ represents the probability assigned by the reference model.
>
> 1. **Explicit Entropy Bonus**: Adding an entropy regularization term $+ \alpha H$ yields a logit gradient $\frac{\partial H}{\partial z_k} = \pi_k(-\log \pi_k - H)$. This forces a positive logit update for any token $k$ satisfying $\pi_k < \exp(-H)$. However, in numerical generation, many incorrect digit tokens can also satisfy this condition. As a result, the entropy bonus spreads probability mass in a largely non-selective way: it promotes diversity, but does not tell the model which token is numerically preferable. In continuous regression, where small token changes can induce very different numerical errors, such undirected exploration is much less effective than targeted guidance toward the error-minimizing token.
>
> 2. **KL Penalty**: Minimizing $D_{\text{KL}}(\pi \| \pi_{\text{ref}})$ produces a logit update direction proportional to $\pi_k \left( \log \frac{\pi_{\text{ref},k}}{\pi_k} + D_{\text{KL}} \right)$. It elevates a token $k$ only if $\pi_k < \pi_{\text{ref},k} \exp(D_{\text{KL}})$. This regularizer encourages the policy to stay close to the reference model rather than directly improving numerical accuracy. Consequently, its effect depends on how informative the reference distribution is for the regression task. If the base model is not numerically precise, its distribution may assign nontrivial probability to multiple suboptimal digits, and the KL penalty will preserve this uncertainty instead of sharpening the policy toward the correct numerical neighborhood. Therefore, while KL regularization can improve stability, it mainly acts as a prior-preserving constraint, not as a targeted correction signal.
>
> In summary, standard stabilizers improve optimization only in a generic sense: the entropy bonus promotes non-selective exploration, while the KL penalty anchors the policy to the reference model. Neither directly identifies the token that best reduces the regression error. In contrast, GenRe² provides a targeted correction signal by explicitly favoring the token that minimizes the continuous numerical loss at each step. This not only prevents mode collapse, but does so in a way that is directly aligned with the regression objective, which explains its consistent advantage over generic RL regularizers in our experiments.
>
> ### Q3 Detailed computational time
>
> Please refer to Q1 of Reviewer AgAT due to space limitations.
>
> ---
>
> **We hope our response has addressed your concerns. If we have missed anything, please let us know.**

---

> > ### Author Rebuttal · Reviewer_MNE6 · 2026-04-03
> >
> > I thank the authors for their comprehensive response and for providing the additional experimental results. Most of my concerns regarding the theoretical mechanism and the normalization issues have been addressed. I will maintain my current score.

---

> > > ### Author Response · Authors · 2026-04-05
> > >
> > > Dear Reviewer MNE6,
> > >
> > > Thanks for your feedback! We are glad to hear that your concerns have been addressed. We sincerely appreciate the time and effort you have dedicated to reviewing our paper and providing thoughtful and valuable comments.
> > >
> > > Best regards
> > >
> > > Authors

---

### Official Review · Reviewer_AgAT · 2026-03-12

**Soundness:** 3
**Presentation:** 3
**Significance:** 3
**Originality:** 3
**Overall Recommendation:** 5
**Confidence:** 3

**Summary:**

This paper considers the problem of using LLMs for regression tasks.

Existing methods for LLM based regression simply apply cross entropy style losses to individual digit tokens. These methods treat each digit equally, however an error in the hundred's digit can be more costly than an error in the one's digit (if the hundred's and ten's digits are correct).

This paper proposes a simple solution to remedy this -- Instead of supervising every single digit separately, generate all digits at once, and simply use RL with terminal supervision. The terminal supervision can be provided, for eg. by the MSE loss between the predicted output and the ground truth value.

**Compliance With Llm Reviewing Policy:**

Affirmed.

**Final Justification:**

The rebuttal addressed my concerns, so I am happy to maintain the positive score.

**Key Questions For Authors:**

1. Using RL for learning regression instead of CE could potentially increase the computation time, especially if the regression targets need to be high precision and have many digits.  It could be useful to have a discussion on this aspect.

**Limitations:**

yes

**Strengths And Weaknesses:**

Strengths.

1. The problem seems timely and important.

2. The proposed remedy to uniform weighting of digits is clean and conceptually simple.

3. The paper is well written and easy to follow.

4. The method shows clear gains in performance over existing baselines, and the evaluation suite is very strong and diverse.

Weaknesses.

1. Using RL for learning regression instead of CE could potentially increase the computation time, especially if the regression targets need to be high precision and have many digits.  It could be useful to have a discussion on this aspect.

---

> ### Author Rebuttal · Authors · 2026-03-31
>
> Thanks for your encouraging feedback! The additional results can be found at can be found at https://anonymous.4open.science/api/repo/GenRe-re-3810/file/AgAT.pdf. Below please find our response.
>
> ### Q1 Discussion on computation time
>
> We agree that training time is an important practical consideration, especially for targets with high precision (many digits). We analyze the computation overhead from two perspectives:
>
> 1. Training cost is inherent to the RL paradigm: To report the training overhead, Table R1-R3 compares the training time between the standard CE baseline and our RL framework. We emphasize that this increased computational cost is strictly limited to the training phase and is inherently driven by the RL paradigm itself (specifically the autoregressive rollouts required for exploration), rather than our proposed modifications. During inference, our model shares the exact same architecture and decoding speed as standard CE models. Therefore, there is zero additional computational burden during deployment, regardless of the target's precision.
>
> 2. Empirical wall-clock time alignment: To investigate whether our performance gains simply stem from increased training compute, we conducted a wall-clock time-aligned experiment. As shown in Table R4, we extended the baselines' (i.e., NTL and DIST²) training to match the exact computational budget and time of the RL framework. Even under the matched time budget, the baselines' performances remain significantly lower than RL-based methods.
>
> This empirically demonstrates that for regression tasks, simply investing more computation time into token level baselines does not close the performance gap. The fundamental advantage stems from the RL formulation itself.
>
> We will include a dedicated discussion on this aspect in the revision as your suggestion. Thank you very much.
>
> ---
>
> **We hope that our response has addressed your concerns, but if we missed anything, please let us know.**

---

> > ### Author Rebuttal · Reviewer_AgAT · 2026-04-03
> >
> > Thanks for the discussion regarding computation time. I am happy to maintain my positive score.

---

> > > ### Author Response · Authors · 2026-04-05
> > >
> > > Dear Reviewer AgAT,
> > >
> > > Thanks for your feedback! We are glad to hear that your concerns have been addressed. We sincerely appreciate the time and effort you have dedicated to reviewing our paper and providing thoughtful and valuable comments.
> > >
> > > Best regards
> > >
> > > Authors

---

### Official Review · Reviewer_9BaU · 2026-03-17

**Soundness:** 3
**Presentation:** 3
**Significance:** 3
**Originality:** 2
**Overall Recommendation:** 3
**Confidence:** 3

**Summary:**

This paper addresses regression tasks reformulated as sequence decoding problems. The authors identify a fundamental limitation of supervised training: it fails to capture the signal from the fully decoded result. Specifically, the numerical gap between the predicted value and the ground truth. Cross-entropy loss treats errors at each token position equally, but in practice, an error in a high-place token is numerically very different from an error in a low-place token. Based on this observation, the paper naturally proposes using reinforcement learning to compensate for this missing signal, enabling the model to perceive the numerical distance between its final prediction and the target value.

The main technical contribution lies in the design choices for RL algorithms. The authors point out that using the default GRPO setting, which normalizes advantages with standard deviation, is problematic for regression tasks, as the absolute scale of the error itself carries important information. They further propose GenRe²: to address the sparse reward and credit assignment issues in pure RL, this method introduces token-level expert supervision by adding a behavior cloning term to the RL objective, providing immediate feedback at each decoding step.

**Compliance With Llm Reviewing Policy:**

Affirmed.

**Final Justification:**

The techniques proposed by the authors are relatively incremental, and the experimental results do not show sufficiently large improvements. Therefore, I will keep the current score.

1. On the issue of removing standard deviation in regression tasks to preserve scale, the approach seems quite intuitive and perhaps too straightforward to warrant a full paper discussion.
2. Regarding the techniques for stabilizing RL training, I appreciate that the authors' response addresses some of my concerns. Figure R3 helps illustrate the advantage of using an expert oracle over ground truth to avoid excessive entropy shrinkage. Still, I note that in Table R1, the performance gap compared to using ground truth remains not very significant.

**Key Questions For Authors:**

1.Compared to ReMax, GenRE's main addition is an extra cross-entropy loss to prevent mode collapse when using only the RL loss. I think an ablation study is needed to compare "cross-entropy with expert oracle," "cross-entropy with original ground truth," and "KL with a reference model." In terms of preventing mode collapse, what specific advantages does the expert offer over the others?

2.Regarding the proposed "cross-entropy with expert oracle":

a. If the prefix predicted so far is correct, does the method use the ground-truth next token?

b. The paper mentions using "TRACT" to provide the oracle signal. How exactly does this work? Does it involve feeding the prefix into a TRACT model, which then predicts the token as the oracle? How is it guaranteed that the expert "minimizes the difference from the ground truth"? How is the prediction bias of TRACT itself addressed?

c. Furthermore, if the oracle minimizes the difference from the ground truth, wouldn't this lead to using more boundary tokens as supervision signals? Would the model predict boundary digits (e.g., 0 or 9) more frequently?

**Limitations:**

yes

**Strengths And Weaknesses:**

Strengths

1.The paper is clearly written and easy to follow. The proposed approach is intuitively reasonable: using episode-level RL supervision to address the misalignment between token-level supervised learning and the overall regression loss.

Weaknesses

1.The two RL training techniques proposed are more engineering-oriented and do not introduce novel theoretical or technical contributions. The issue of bias introduced by normalizing advantages with standard deviation has already been discussed in many RLVR papers (e.g., Dr. GRPO [1]), and removing this term for regression tasks is a fairly intuitive fix.

2.Similarly, using cross-entropy or behavior cloning losses alongside RL optimization is a common regularization trick (e.g., PPO-ptx in [2]).

3.The proposed "cross-entropy with expert oracle" requires further explanation and experimental support to justify its necessity and rationale. See questions below.

[1] Liu, Zichen, et al. "Understanding r1-zero-like training: A critical perspective." arXiv preprint arXiv:2503.20783 (2025).

[2] Ouyang, Long, et al. "Training language models to follow instructions with human feedback." Advances in neural information processing systems 35 (2022): 27730-27744.

---

> ### Author Rebuttal · Authors · 2026-03-31
>
> Thanks for your valuable comments. The results can be found at https://anonymous.4open.science/api/repo/GenRe-re-3810/file/9BaU.pdf. Below please find our response.
>
> ### W1 Distinction from Dr.GRPO
>
> Thanks for pointing this out. We agree the two methods **look similar** on the surface because both remove the standard-deviation normalization (std norm.) term. However, their **underlying mechanisms** differ: they target different failure modes under different reward regimes.
>
> 1. The task setting is different. Dr.GRPO focuses on RLVR with mostly binary rewards, while our setting is decoding-based regression with continuous rewards (e.g., negative MSE). Thus, the key supervision signal is absolute error magnitude, not only relative ranking within a rollout group.
>
> 2. The role of std norm. is different. In Dr.GRPO, removing std norm. mainly addresses unstable or overly large weights when within-group variance is near zero. In our regression setting, the dominant issue is that std norm. erases absolute-scale information. For example, reward groups `[-0.1,-0.2,-0.3]` and `[-100,-200,-300]` yield identical normalized advantages after std norm, although the second group corresponds to catastrophic errors. This forces similarly scaled updates for minor and severe errors, weakening error-severity discrimination.
>
> 3. Our claim is supported by targeted falsification rather than intuition. To test whether the degradation is merely small-std instability, we run three ablations: (1) tanh-bounded normalized advantages, (2) gradient norm., and (3) binary 0-1 rewards that remove scale and naturally avoid advantage explosion. As shown in Fig. R1-R2, none recovers the performance of removing std norm. This indicates that the key issue in decoding-based regression is the loss of absolute error information, not just unstable weighting.
>
> In short, our contribution is not a generic engineering tweak, but the identification and validation of a regression-specific failure mode of standard GRPO-style norm.. We will clarify this distinction and include the new ablation results in the revision. Thank you.
>
>
> ### W2 Distinction from PPO-ptx
>
> Thanks for pointing this out. Unlike standard CE regularization (e.g., PPO-ptx), which supervises only the ground-truth trajectory and thus offers little correction for off-trajectory rollout states, GenRe² provides dynamic, state-aware guidance. At each rollout position, GenRe² computes the error-minimizing next token, yielding step-wise recovery signals even after deviation from the GT path.
>
> ### Q1 Comparison with other regularization
>
> Table R1 compares GenRe² (CE with expert oracle) with CE using GT tokens and CE with KL regularization, where our GenRe² performs best.
>
> We also analyze the entropy dynamics to explain this gain. With entropy $H$, logit $z_k$, and token probability $\pi_k$, the gradient is $\frac{\partial H}{\partial z_k} = \pi_k(-\log \pi_k - H)$. Increasing a token's probability raises entropy only when $\pi_k < \exp(-H)$.
>
> GT tokens usually have high prior probability and often fail this condition. When generation deviates, GenRe² instead selects an optimal low-probability token that is more likely to increase entropy while correcting error. Figure R3 confirms that GenRe² targets these entropy-increasing tokens, unlike CE with GT.
>
> Therefore, GenRe² injects entropy to mitigate mode collapse while directly optimizing continuous regression error. For detailed comparison with KL regularization, please see our response to Reviewer MNE6 (Q2). We will add these results and discussions in the revision.
>
> ### Q2a If the prefix is correct, using the ground-truth next token?
>
> Yes. For tabular and code regression tasks, if the generated prefix perfectly matches the ground-truth prefix, the oracle deterministically outputs the ground-truth next token.
>
> ### Q2b TRACT as oracle?
>
> Thanks for the comment. We clarify that for tabular and code-metric regression, outputs are number-structured, so we use a deterministic mathematical oracle directly, not TRACT. By contrast, GRM tasks contain free-form CoT text that cannot be supervised by a deterministic oracle. Thus we use TRACT as a reference model and apply a KL penalty to its output distribution for step-wise guidance (as noted in lines 369-371), similar to on-policy distillation. While this distillation cannot remove TRACT's bias, we assume TRACT is a capable expert; Table R2 further shows that stronger teachers consistently yield stronger GenRe² models, confirming effective capability transfer.
>
> ### Q2c Predict boundary more?
>
> Yes. Fig. R4 shows that GenRe² predicts more boundary. The increase is modest when the first decoded token is correct, but large after a first-token error, indicating that oracle-guided supervision uses boundary tokens mainly for post-prefix correction rather than creating a general boundary-token bias.
>
> ---
>
> **We hope that our response has addressed your concerns, but if we missed anything, please let us know.**

---

### Official Review · Reviewer_cFw2 · 2026-03-18

**Soundness:** 3
**Presentation:** 3
**Significance:** 3
**Originality:** 3
**Overall Recommendation:** 3
**Confidence:** 4

**Summary:**

The paper addresses the mismatch between token-level cross-entropy loss and continuous target values in decoding-based regression. The authors formulate the generation process as a Markov Decision Process and use sequence-level rewards optimized via the ReMax algorithm. To mitigate the sparse reward problem and mode collapse, they introduce an on-policy token-level expert guidance term. Evaluations on tabular data, code metrics, and generative reward modeling show that this combined approach outperforms standard token-level losses.

**Compliance With Llm Reviewing Policy:**

Affirmed.

**Final Justification:**

The rebuttal addresses some of my concerns, but important issues remain, especially regarding the practical oracle implementation for complex tokenizations and the robustness of the method against catastrophic outliers during generation. Therefore, I maintain my score.

**Key Questions For Authors:**

See weaknesses.

**Limitations:**

yes

**Strengths And Weaknesses:**

Strengths

(1) The observation that token-level distance penalties operate locally and overlook cumulative sequence-level errors is highly accurate. Formulating decoding-based regression as a sequence-level optimization problem inherently aligns the training objective with the true regression metrics.

(2) Section 3.4 provides a very convincing mathematical explanation for why ReMax is better suited for regression than GRPO. The derivation of $A_{ReMax}=\left(E_{greedy}-E_{sample}\right)\cdot\left(E_{greedy}+E_{sample}\right)$  clearly illustrates how it preserves the absolute error scale, avoiding the scale invariance caused by GRPO's standard deviation normalization.

(3) The paper logically connects the degradation of high-order tokens to the sparse nature of RL rewards. Integrating dense token-level expert feedback effectively acts as a regularizer, successfully preventing the drastic entropy drop and premature convergence typically associated with pure RL.

(4) The empirical evaluation is extensive. Testing the method across tabular benchmarks, code metrics, and reward modeling, while averaging over 5 random seeds, provides strong evidence for the consistency of the results.

Weaknesses

(1) 	Page 6, Section 3.5, the proposed GenRe^2\ relies on an on-policy expert oracle $\pi^\ast$ that dynamically determines the optimal next token ${\ a}_t^\ast$ to minimize the distance to the target $y$ given the current trajectory ${\hat{y}}_{<t}$. While this approach works cleanly for simple integer tokenizations, the manuscript lacks critical algorithmic details on how this oracle is implemented for complex, continuous-space tokenizations. For such highly structured formats, analytically determining the single optimal next token that minimizes the final sequence-level MSE requires either exhaustive search over all valid completions or a complex inverse-mapping algorithm. The paper need to explicitly formalizing the oracle's construction for these advanced tokenizers and discussing the associated computational complexity, as evaluating this dynamically at every generation step during RL rollouts could introduce severe computational bottlenecks.

(2) 	Appendix F.4 states the rollout phase generates 16 samples per input for Tabular tasks and 4 for Code Metric tasks. Combined with the requirement to compute token-level expert guidance at every generation step, this setup inherently introduces massive computational and memory overheads during training. However, the manuscript completely omits any empirical comparison of training costs against baseline methods. A fair evaluation of training efficiency is crucial for assessing the practical utility of this RL framework compared to purely supervised methods like $NTL $ or $DIST^2$.

(3) 	Although the paper employs an outlier filtering strategy for the generated candidates to recover positive $R^2$ scores, the authors explicitly acknowledge that $ GenRe^2$ cannot eliminate the hallucinations inherent in unbounded tokenization. Since the core motivation of introducing sequence-level RL is to equip the model with global magnitude awareness, it remains concerning that the policy still produces such extreme outliers before filtering that a mean $R^2 $ of -$4.44\times10^{17} $ for P10. The paper needs to further explore why the sequence-level reward fails to constrain these catastrophic bounds during the generation phase itself.

(4) Minor Presentation Issue: In Section 3.5, the text incorrectly references “Figure 3.3 presents the results”, which should clearly be Figure 4.

---

> ### Author Rebuttal · Authors · 2026-03-31
>
> Thanks for your valuable comments. The additional results can be found at  https://anonymous.4open.science/api/repo/GenRe-re-3810/file/cFw2.pdf. Below please find our response.
>
> ### W1 Oracle construction and computational complexity
>
> We will explicitly formalize the deterministic oracle construction in the revision to clarify that it introduces no computational bottleneck, for two key reasons:
>
> First, the oracle is not evaluated dynamically during generation. Model rollouts proceed normally without interruption. The expert targets are computed entirely after the rollout finishes, adding zero overhead to the generation speed.
>
> Second, given a generated prefix and the target value, the ground-truth token is determined via constant-time $\mathcal{O}(1)$ mathematical boundary evaluations. This process eliminates the need for expensive search trees or additional model inference, ensuring high computational efficiency.
>
> Under the sign-mantissa-exponent sequence of P10, given a correct positive sign (+) and a first mantissa digit exceeding the ground truth (GT), the oracle evaluates two deterministic boundary completions: (i) padding the remaining mantissa with '9's while decrementing the exponent, versus (ii) padding with '0's while maintaining the original exponent. The completion minimizing the absolute distance to the GT is instantly selected to determine the target token (0 or 9). This deterministic logic consistently governs other digit positions and alternative tokenization schemes.
>
> Given any generated prefix, determining the optimal next token relies strictly on basic arithmetic comparisons across at most two boundary conditions, ensuring an $\mathcal{O}(1)$ decision complexity.
> This direct mathematical mapping introduces negligible computational overhead, effectively preventing training bottlenecks.
> Comprehensive algorithmic workflows detailing the oracle generation process across different tokenizers will be included in our revised paper.
>
> ### W2 Training costs and computational overhead comparison
>
> Thank you for this important comment.
>
> First, the additional overhead mainly comes from autoregressive exploration in RL fine-tuning (e.g., generating 16 samples per rollout). By contrast, token-level expert scoring is an $\mathcal{O}(1)$ post-rollout operation and does not increase generation-time complexity.
>
> To quantify this overhead, Tables R1-R3 compare memory usage and wall-clock time per training step (or epoch) between the supervised baseline and our RL framework. As expected, RL consumes more resources due to parallel rollouts.
>
> For a fair comparison, we also run a wall-clock-aligned experiment by extending supervised training to match the total time budget of our RL framework.
>
> As shown in Table R4, under matched time budgets, the supervised baseline plateaus and remains below our RL framework. This result indicates that the gain primarily comes from sequence-level RL optimization for credit assignment, rather than from simply using more compute.
>
> We will include a dedicated training-cost comparison in the revision. Thank you.
>
> ### W3 Catastrophic bounds and sequence-level rewards
>
> We thank the reviewer for this insightful question. Catastrophic errors are a well-known issue in standard numerical tokenization. As noted by [1], encodings such as P10 naturally "yield a small percentage of highly erroneous predictions.'' The reason is straightforward: a single mistake in a high-order digit can cause a very large error in the decoded value.
>
> During training, standard CE maximizes only the exact ground-truth token. It assigns the same penalty to all incorrect tokens and therefore does not distinguish a small decimal deviation from a catastrophic high-order error. By contrast, sequence-level RL (e.g., ReMax) optimizes final continuous error directly. It penalizes catastrophic outcomes much more strongly, explicitly reducing the probability of risky high-order token choices.
>
> That said, fully eliminating such outliers at generation time remains difficult, especially on unseen test inputs. In a discrete token space, even mild predictive uncertainty can still trigger an incorrect high-order token.
>
> To quantify mitigation quality, we report the proportion of catastrophic outliers in Table R5. The outlier ratio consistently follows CE $>$ ReMax $>$ GenRe². This pattern shows that sequence-level RL substantially reduces catastrophic outliers relative to CE, and that our Expert Oracle provides the strongest suppression. We will add this discussion in the paper. Thank you very much.
>
> [1] xVal: A Continuous Numerical Tokenization for Scientific Language Models. NeurIPS Workshop, 2023.
>
> ### W4 Typo
>
> Thanks for pointing these out. We have revised them and checked thoroughly. Thank you very much.
>
> ---
>
> **We hope that our response has addressed your concerns, but if we missed anything, please let us know.**

---

> > ### Author Rebuttal · Reviewer_cFw2 · 2026-04-01
> >
> > The authors have adequately addressed my main concerns raised in the original review. In particular, their rebuttal clarifies the technical points I was uncertain about and provides sufficient explanations regarding the experimental/theoretical issues. Based on these clarifications, my previous concerns are fully resolved.

---

> > > ### Author Response · Authors · 2026-04-02
> > >
> > > Dear Reviewer cFw2,
> > >
> > > Thank you for your timely response and for acknowledging that your main concerns have been fully resolved.
> > >
> > > We noticed that you kindly selected the option: "*(a) Fully resolved - My concerns have been adequately addressed. If you select this option, please consider adjusting your score accordingly*" and commented that your previous concerns are fully resolved. However, we observed that your score has not yet been updated.
> > >
> > > May we politely ask if there are still any other remaining concerns that we have not addressed? If you have any further questions or require additional clarification, we would be more than happy to continue the discussion.
> > >
> > > Thank you again for your thoughtful comments and the time you have dedicated to reviewing our work.
> > >
> > > Best regards,
> > >
> > > Authors

---

### Official Review · Reviewer_aPNj · 2026-03-20

**Soundness:** 3
**Presentation:** 4
**Significance:** 3
**Originality:** 3
**Overall Recommendation:** 5
**Confidence:** 3

**Summary:**

The authors propose a new method that uses sequence-level RL to be aware of the sequence-level numerical magnitude in regression. They also identify ReAMax as a more suitable RL algorithm for regression. Experiments on three tasks demonstrate the effectiveness of the method.

**Compliance With Llm Reviewing Policy:**

Affirmed.

**Final Justification:**

The weaknesses are largely addressed. I would also appreciate if the ablations can be incorporated in revised versions in more settings. I have increased my score during the rebuttal phase and lean toward acceptance, given the novelty of the algorithm and the comprehensive support for its claims.

**Key Questions For Authors:**

Figure 3 can be misleading: For GRPO w/o std, it only shows that it is more prone to have smaller (vanishing) gradient norms regardless of reward scale (most blue points are around the x-axis), while it does not effectively show that high-reward samples (small errors) yield vanishing gradients, allowing the model to settle (row 270–271, left column).

Please refer to the cons for the rest of the questions.

**Limitations:**

Please refer to the cons and questions.

**Strengths And Weaknesses:**

**Pros:**

1. Novel point of addressing decoding-based regression using a sequence-level objective to capture the global magnitude of the target value.
2. Compare two types of popular RL algorithms and discover a more appropriate RL algorithm due to the uniqueness of the regression problem.
3. Good writing that delivers the story smoothly.

**Cons:**

1. Effectiveness of token-level supervision: In rows 156–158 (right column), the authors criticize *uniform weighting (of CE), claiming it fails to prioritize critical high-bit information, causing the token-level surrogate loss to deviate from the true regression objective*. However, the token-level correction added (row 311–313) via behavior cloning seems to give uniform weights to each token. Also, would non-uniform but decreasing weights, which consider the magnitude information in each token (i.e., the larger the index for the generated token, the smaller the weight), perform also decently?
    - Additionally, the claim that the “*policy struggles to distinguish between a fundamental structural mistake (wrong magnitude) and a minor precision error*” (row 256–258, right column) can be misleading. The difference between structural mistakes and minor precision shall be reflected in the reward indirectly (a larger reward means more structural mistakes).
2. Inconsistent model architectures (NTL consistently uses T5) across the three tasks, and the tasks seem to be different from the NTL baseline.

---

> ### Author Rebuttal · Authors · 2026-03-31
>
> Thanks for your valuable comments. The additional results can be found at https://anonymous.4open.science/api/repo/GenRe-re-3810/file/aPNj.pdf. Below please find our response.
>
> ### W1 Effectiveness of non-uniform decreasing weights
>
> Thanks for the insightful suggestion. We conducted ablation experiments applying decreasing weights (where a larger index yields a smaller weight) to both the standard CE loss and our token-level on-policy correction. Specifically, to ensure a fair comparison with CE, we implement a smoothed exponential weighting scheme using $\log(\text{base}^k+1)$ (where $k$ represents the digit's positional significance). These weights are then normalized and multiplied by the total sequence length to guarantee that the overall gradient update magnitude remains identical to the uniform CE loss.
>
> As shown in Table R1, while reweighting indeed brings marginal improvements over uniform weighting, the overall gains are limited and smaller than GenRe². This reinforces our core premise: while localized, non-uniform token weighting helps slightly, the primary bottleneck in decoding-based regression is the lack of global, sequence-level awareness. The significant performance leap achieved by GenRe² confirms that sequence-level RL optimization is the crucial factor for resolving this bottleneck.
>
> We will include these results in the revised paper. Thank you.
>
> ### W1-1 Clarification on "structural mistakes" and reward reflection
>
> We apologize for the ambiguity in lines 256-258. We fully agree that the sequence-level scalar reward indirectly reflects the severity of the mistake through its magnitude. Our intended argument was specifically regarding the *temporal credit assignment issue* inherent in RL. A large negative terminal reward tells the model that the final prediction is bad, but not which token-level choice caused the error or how to correct it. By introducing dense token-level supervision, we provide exact, step-wise directional guidance.
>
> We will revise the paper to explicitly frame this discussion in detail. Thank you very much.
>
> ### W2 Inconsistent model architectures and tasks compared to NTL
>
> We thank the reviewer for this constructive suggestion. We deliberately tailored the model architectures to best suit each specific task, closely following established choices in related prior works [1-3]. To ensure a rigorous and direct comparison with NTL, we have conducted an additional experiment strictly following the exact task setting and T5 architecture from Table A3 of the original NTL paper.
>
> Table R2 confirms that NTL brings a substantial improvement over the standard CE baseline, demonstrating its strong effectiveness. Furthermore, GenRe² achieves an additional yet consistent performance gain over NTL.
>
> We attribute this performance gap to the fundamental difference between static and dynamic learning mechanisms. While NTL elegantly aligns numerical representations based on static ground-truth targets, GenRe² allows the model to learn dynamically from its on-policy rollouts during training. By actively exploring the action space,  on-policy RL learns to self-correct the specific deviations it is naturally prone to make during inference, thereby showing better generalization ability [4].
>
> We will include more experiments related to the original NTL paper in the revised version. Thank you very much.
>
> [1] Decoding-based Regression. TMLR 2025
>
> [2] Regression Language Models for Code. arXiv, 2025
>
> [3] TRACT: Regression-Aware Fine-tuning Meets Chain-of-Thought Reasoning for LLM-as-a-Judge. ACL 2025.
>
> [4] Retaining by Doing: The Role of On-Policy Data in Mitigating Forgetting. arXiv 2025.
>
> ### Q1 Clarification on Figure 3 and vanishing gradients
>
> We appreciate your careful observation. We acknowledge that the original Figure 3 was visually misleading due to scale. Our point is not an absolute gap in gradient magnitude, but the alignment between reward and update signal. We observe a clear trend: higher-reward samples generally have smaller gradients that support convergence, while lower-reward samples produce relatively larger updates that guide optimization. To illustrate this more clearly, we rescaled the plot and tracked per-sample gradient norms across checkpoints and training epochs. As shown in Figure R1, the revised plots demonstrate that low-reward samples do not suffer vanishing gradients and still provide healthy updates. We will replace the original Figure 3 with these clearer rescaled visualizations in the revised paper. Thank you very much.
>
> ---
>
> **We hope that our response has addressed your concerns, but if we missed anything, please let us know.**

---

> > ### Author Rebuttal · Reviewer_aPNj · 2026-04-03
> >
> > Thanks to the authors for the additional experiments and more comprehensive explanations. Most of my concerns are resolved. One minor point is that the ablations are conducted on a single dataset; it would be better if they could be evaluated across more settings. Additionally, the CE ablation is performed on the 100 TALENT regression task, whereas the original paper for this setting does not include CE, making this choice somewhat confusing and less natural. However, I would like to improve opinions because the majority of concerns are addressed.

---

> > > ### Author Response · Authors · 2026-04-04
> > >
> > > Dear Reviewer aPNj,
> > >
> > > Thanks for your timely and constructive response! We are glad to hear that our additional experiments have resolved the majority of your concerns. All newly added tables mentioned below can be found at https://anonymous.4open.science/api/repo/GenRe-re-3810/file/aPNjR2.pdf.
> > >
> > > Following your valuable advice, we are actively expanding our ablation studies across broader settings beyond the TALENT benchmark. As part of this ongoing effort, we have explicitly incorporated new baselines, specifically Reweight CE + NTL and Reweight CE + Reweight NTL. The results for these newly added variants are presented in Table RR1 and Table RR2.
> > >
> > > Regarding the CE ablation on the TALENT regression task, we would like to clarify our original setup. In this tabular regression setting, the base model is inherently optimized via standard CE, which is why a standalone "CE" row was initially omitted. We apologize if this omission caused any confusion. To ensure a rigorous and natural comparison, we have now explicitly evaluated this CE ablation. The preliminary results are provided in Table RR3. We are currently running the remaining random seeds and will include the complete evaluation in the final revision.
> > >
> > > Thank you again for your constructive feedback and for recognizing our efforts! We will include this discussion in the revised paper. Thank you very much.

---

### Decision · Program_Chairs · 2026-04-30

**Decision:**

Accept (regular)

**Comment:**

After considering the reviews, author rebuttal, and discussion, I recommend acceptance.

Reviewers broadly agree that the paper addresses a meaningful limitation of decoding-based regression, namely the mismatch between token-level supervision and sequence-level numerical error, and that the proposed formulation is technically sound and clearly presented. Several reviewers also found the empirical evaluation to be strong, with consistent gains across multiple tasks and settings. In particular, the paper’s main strengths are the sequence-level formulation for numerical prediction, the analysis of RL design choices for regression, and the breadth of experiments.

The main concerns raised during review were about novelty, the construction and cost of the expert oracle, comparisons to alternative RL normalization and stabilization strategies, and computational overhead. Based on the rebuttal and follow-up discussion, these concerns were substantially addressed for most reviewers. In particular, the rebuttal clarified the oracle design and its cost, added discussion and experiments regarding training overhead and wall-clock comparisons, and addressed questions about alternative normalization and regularization choices. Reviewers aPNj, AgAT, and MNE6 remained positive after rebuttal, and reviewer cFw2 explicitly stated that their main concerns were fully resolved, even though their score was not updated.

A remaining concern, as mentioned by reviewer 9BaU, is that some components may appear incremental relative to prior RL practice, and the gains of oracle supervision over simpler alternatives are not always large in the additional ablations. I agree that the paper should moderate any overly broad novelty claims and present these distinctions carefully in the final version. That said, I do not find this concern sufficient to overturn the stronger overall case for acceptance, given the paper’s soundness, clarity, and solid empirical support.

Overall, this is a well-executed and useful contribution on decoding-based regression. For the camera-ready version, the authors should make sure to incorporate the rebuttal clarifications, especially on oracle construction, computational overhead, and the precise scope of the novelty claims.